# Dual-quartet phosphorescent emission in the open-shell $M_1Ag_{13}$ (M = Pt, Pd) nanoclusters

Cao Fang[1,2,5], Chang Xu[1,5], Wei Zhang [3], Meng Zhou [3], Dong Tan[1,2], Lixia Qian[1], Daqiao Hu [1,2] ✉, Shan Jin [2,4] ✉ & Manzhou Zhu [1,2] ✉

Dual emission (DE) in nanoclusters (NCs) is considerably significant in the research and application of ratiometric sensing, bioimaging, and novel optoelectronic devices. Exploring the DE mechanism in open-shell NCs with doublet or quartet emissions remains challenging because synthesizing open-shell NCs is difficult due to their inherent instability. Here, we synthesize two dual-emissive $M_1Ag_{13}(PFBT)_6(TPP)_7$ (M = Pt, Pd; PFBT = penta-fluorobenzenethiol; TPP = triphenylphosphine) NCs with a 7-electron open-shell configuration to reveal the DE mechanism. Both NCs comprise a crown-like $M_1Ag_{11}$ kernel with Pt or Pd in the center surrounded by five $PPh_3$ ligands and two $Ag(SR)_3(PPh_3)$ motifs. The combined experimental and theoretical studies revealed the origin of DE in $Pt_1Ag_{13}$ and $Pd_1Ag_{13}$. Specifically, the high-energy visible emission and the low-energy near-infrared emission arise from two distinct quartet excited states: the core-shell charge transfer and core-based states, respectively. Moreover, PFBT ligands are found to play an important role in the existence of DE, as its low-lying $\pi^*$ levels result in energetically accessible core-shell transitions. This novel report on the dual-quartet phosphorescent emission in NCs with an open-shell electronic configuration advances insights into the origin of dual-emissive NCs and promotes their potential application in magnetoluminescence and novel optoelectronic devices.

Luminescent nanoclusters (NCs) with atomic precision (1–3 nm in diameter) have shown great potential in various fields, such as optical waveguides[1], light-emitting diodes[2–6], and bioimaging[7–14]. In particular, dual-emitting NCs with superior accuracy and ratio metric analyzes may become more prevalent than expected[15–19]. Nevertheless, the synthesis of dual-emitting NCs remains a formidable challenge because it requires precise synthesis and a deep understanding of the dual emission (DE) property-structure correlation[20–25]. Atomically precise NCs with exact atomic configurations are ideal platforms for investigating photoluminescence (PL) mechanisms, which enables

customizing the PL properties of these NCs[26–28]. Generally, the origin of DE in NCs can be ascribed to two first excited singlet states ($S_1$), $S_1$ and the lowest triplet excited state ($T_1$), and two $T_1$ states based on recent research on DE mechanisms of Au or AuCu NCs[29–32]. First, the DE may originate from the two equilibrium configurations of the $S_1$ in one NC. As reported by Jin et al., the presence of two equilibrium configurations of the $S_1$ state is primarily attributed to structural distortion accompanied by electron redistribution in the dual-emitting $Au_{24}$ NC and other Au nanoclusters[29]. Second, the origin of DE may be ascribed to the $S_1$ state and the $T_1$. The less-allowed $T_1 \rightarrow S_1$ transition might

[1]Department of Chemistry and Centre for Atomic Engineering of Advanced Materials, Anhui University, Hefei, Anhui 230601, China. [2]Key Laboratory of Structure and Functional Regulation of Hybrid Materials of Ministry of Education, Anhui University, Hefei, Anhui 230601, China. [3]Hefei National Laboratory for Physical Sciences at the Microscale, University of Science and Technology of China, Hefei, Anhui 230026, China. [4]Institutes of Physical Science and Information Technology, Anhui University, Hefei, Anhui 230601, China. [5]These authors contributed equally: Cao Fang, Chang Xu. ✉e-mail: hudaqiao@ahu.edu.cn; jinshan@ahu.edu.cn; zmz@ahu.edu.cn

involve a change in spin multiplicity. For example, Jin et al. discovered that in $Au_{42}(PET)_{32}$ (PET = 2-phenylethanethiolate) NC, DE arises from the $S_1$ and $T_1$ states, with intersystem crossing (ISC) occurring between these two states. Moreover, the ISC rate accelerated when the $Au_{42}$ NC was embedded in the film, induced by dipolar interactions[30]. Similarly, fluorescence ($S_1$ state) and phosphorescence ($T_1$ state) were observed in $Au_2Cu_6$ NC, as reported by Mitsui et al., with thermally activated ISC between $S_2$ and $T_2$ states caused by the spin-vibronic coupling effects[31]. Finally, DE can also originate from two triplet states corresponding to the core and shell-based states. Sun et al. reported a near-infrared (NIR) dual-phosphorescent $Au_{20}$ NC that originates from ligand-to-kernel and kernel-based states[32].

Although several studies have been conducted to reveal the PL mechanism of dual-emissive NCs, they are primarily limited to NCs with closed-shell structures which are preferred in NCs according to superatom theory and electron-counting rules[33]. The instability of open-shell NCs with one unpaired electron may restrict its research on PL. Meanwhile, the open-shell molecules emit differently from the closed-shell molecules. Generally, they exhibit doublet fluorescence emission from the first excited doublet state ($D_1$) to the ground state ($D_0$)[34-38]. Quartet phosphorescent emission from the first excited quartet state ($Q_1$) to the $D_0$ state has rarely been reported because of the higher energy level of the $Q_1$ state. Thus, identifying open-shell NCs with unpaired electrons in the excited state may provide an alternative strategy for further understanding the PL mechanism of dual-emissive NCs.

In this work, we report an example of DE in open-shell NCs, namely $Pt_1Ag_{13}(PFBT)_6(TPP)_7$ ($Pt_1Ag_{13}$; PFBT = pentafluorobenzenethiol; TPP = triphenylphosphine) and $Pd_1Ag_{13}(PFBT)_6(TPP)_7$ ($Pd_1Ag_{13}$). The two NCs possess a crown-like $M_{12}$ kernel with the loss of one vertex Ag atom icosahedral $M_{13}$ kernel, resulting in 7-electron open-shell configurations. Significantly, $Pt_1Ag_{13}$ and $Pd_1Ag_{13}$ NCs showed dual-quartet phosphorescent emissions, which are rarely reported for open-shell molecules. The experimental results reveal that DE may originate from two emitting states in one NC. Theoretical calculations further demonstrate that DE primarily originates in the core-shell charge transfer (CT) and core-based states. The obtained insights fill the gap in the DE mechanism of Ag-alloyed NCs with an open-shell electron configuration and open avenues for the magnetoluminescence and novel optoelectronic applications of NCs[39,40].

## Results

### Synthesis and characterization of $Pt_1Ag_{13}$ and $Pd_1Ag_{13}$ nanoclusters

$Pt_1Ag_{13}$ and $Pd_1Ag_{13}$ were successfully synthesized using a convenient one-pot synthetic method, as described in the Methods. The overall structure is shown in Fig. 1a, b. The chemical compositions of $Pt_1Ag_{13}$ and $Pd_1Ag_{13}$ were confirmed by electrospray ionization mass spectrometry (ESI-MS) in the positive ion mode. In the ESI-MS spectrum of $Pt_1Ag_{13}$, a strong signal was observed at m/z 4761.07 (Fig. 1c), corresponding to the $[Pt_1Ag_{13}(PFBT)_6(TPP)_7Cs]^+$ species (calculated at m/z 4761.06). Similarly, the ESI-MS spectrum of $Pd_1Ag_{13}$ exhibited a prominent peak at m/z 4673.03, which was assigned to $[Pd_1Ag_{13}(PFBT)_6(TPP)_7Cs]^+$ (calculated at m/z 4673.00) (Fig. 1d). These assignments were supported by the excellent correlation between the experimental and simulated isotopic patterns, indicating that both NCs were charge-neutral. The thermogravimetric measurements (Supplementary Figs. 1, 2) showed a weight loss of 65.21% for $Pt_1Ag_{13}$ and 66.45% for $Pd_1Ag_{13}$, respectively, which aligned with the theoretical ligand contents of 65.48% for $Pt_1Ag_{13}$ and 66.75% for $Pd_1Ag_{13}$, respectively. X-ray photoelectron spectroscopy confirmed the presence of Pt and Ag in $Pt_1Ag_{13}$ and Pd and Ag in $Pd_1Ag_{13}$. Pt and Pd exhibited nearly zero valence states at the centers of $Pt_1Ag_{13}$ or $Pd_1Ag_{13}$ (Supplementary Figs. 3–6). The Ag $3d_{5/2}$ peak at 368.4 eV suggests that the Ag valence state is close to that of Ag(I) (Supplementary Fig. 7). Inductively coupled plasma-atomic emission spectroscopy and energy-dispersive spectrometry (Supplementary Figs. 8, 9 and Supplementary Table 1) were performed to identify the composition and purity of the NCs. Moreover, $Pt_1Ag_{13}$ and $Pd_1Ag_{13}$ exhibited an odd number of valence electrons (13−6 = 7), similar to previously reported $Ag_{23}$ and $Ag_{34}$ NCs[41,42]. The presence of an unpaired electron was further evidenced by a solid-state electron paramagnetic resonance analysis, which showed a strong signal at g = 2.003, with one local maximum and one local minimum (Supplementary Figs. 10, 11).

Single-crystal X-ray crystallography revealed that $Pt_1Ag_{13}$ and $Pd_1Ag_{13}$ adopt the $P–1$ space group (Supplementary Figs. 12, 13 and Supplementary Tables 2, 3). Both NCs comprised a crown-like $M_1Ag_{11}$ kernel with Pt or Pd atoms in the center surrounded by five $PPh_3$ ligands and two $Ag(SR)_3(PPh_3)$ motifs (Fig. 1e). Compared to the icosahedral $M_{13}$ kernel of $Pt_1Ag_{14}$ reported by Huang et al.[43]. the loss of one vertex Ag atom in the crown-like $M_{12}$ kernel caused structural

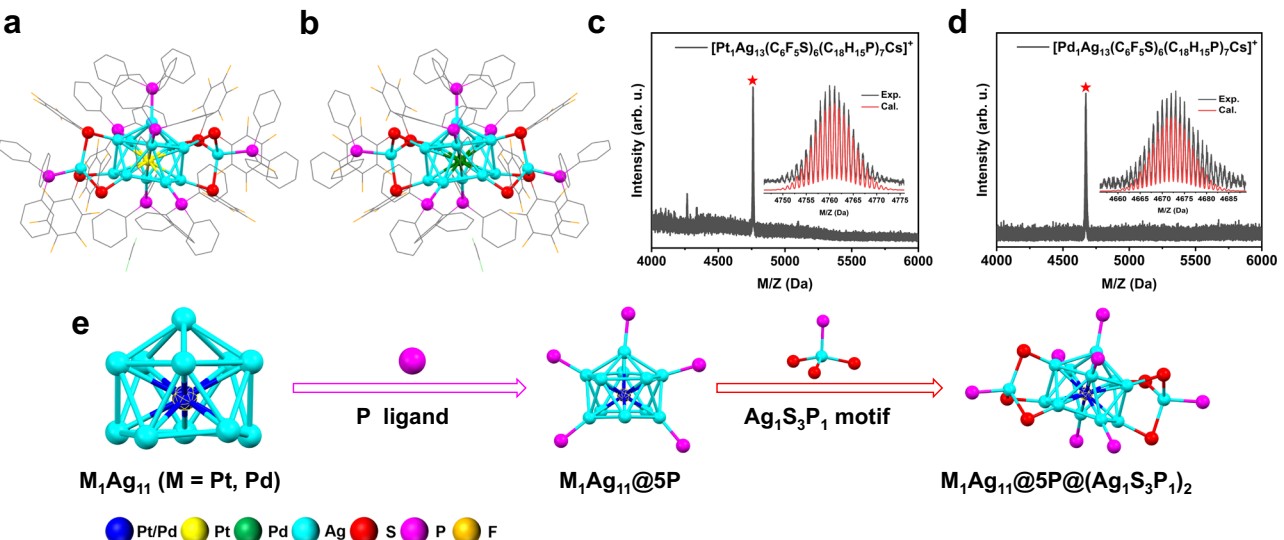

**Fig. 1 | X-ray structure and ESI-MS results of $Pt_1Ag_{13}$ and $Pd_1Ag_{13}$. a, b** Ball-and-stick representation of $Pt_1Ag_{13}$ (**a**) and $Pd_1Ag_{13}$ (**b**). **c, d** ESI-MS results of $Pt_1Ag_{13}$ (**c**) and $Pd_1Ag_{13}$ (**d**) nanoclusters. Insets: experimental (in black) and simulated (in red) isotope patterns. **e** Sequence from inside to outside showing $M_1Ag_{11}$ (M = Pt, Pd) kernel, P ligands, $M_1Ag_{11}@5P$, two $Ag_1S_3P_1$ motifs, and $M_1Ag_{11}@5P@(Ag_1S_3P_1)$ frame. For clarity, the H atoms are omitted. Color labels: blue, Pt or Pd; yellow, Pt; green, Pd; sky blue, Ag; red, S; magenta, P; orange, F; gray, C.

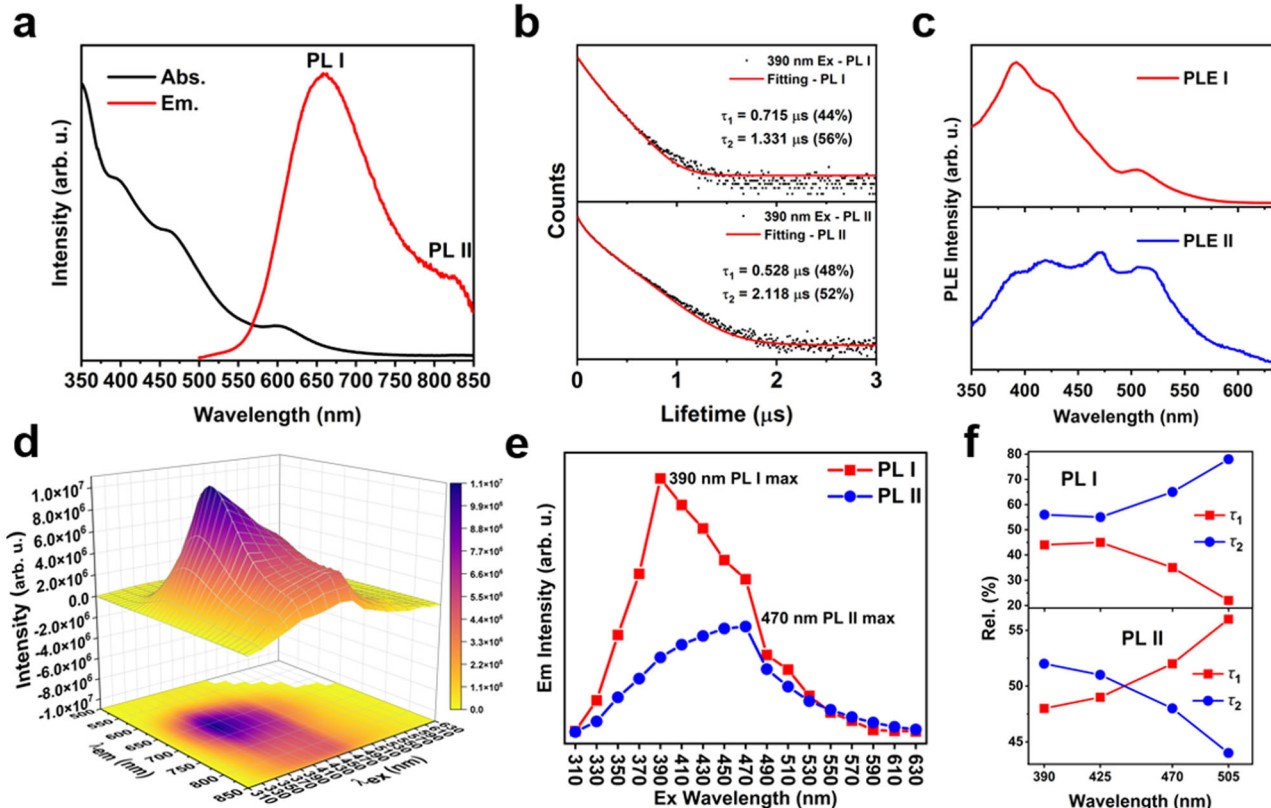

**Fig. 2 | Optical Properties of Pt₁Ag₁₃.** **a** Ultraviolet-visible absorption and photoluminescence (PL) spectra of $Pt_1Ag_{13}$ in 2-Me-THF. Abs: absorption, Em: emission. **b** PL I and PL II decay curves of $Pt_1Ag_{13}$ at 390 nm excitation. Ex: excitation. **c** PL excitation (PLE) spectrum of $Pt_1Ag_{13}$ at PL I and PL II wavelengths. **d** Three-dimensional (3D) consecutive PLE/PL map of $Pt_1Ag_{13}$. $\lambda_{ex}$: excitation wavelength, $\lambda_{em}$: emission wavelength. **e** PL intensity of PL I and PL II of $Pt_1Ag_{13}$ in 2-Me-THF at different excitation wavelengths. **f** The percentage of lifetime $\tau_1$ and $\tau_2$ of PL I and PL II in $Pt_1Ag_{13}$ excited at 390, 425, 470, and 505 nm, respectively.

distortion with five neighboring non-coplanar Ag atoms, changing the Ag-Ag-Ag bond angle (Supplementary Fig. 14). Most metal NCs with $M_{13}$ as the kernel contained complete surface shells and closed electronic configurations. However, it was extremely rare for vacant atomic positions to form on the standard $M_{13}$ core. Therefore, $Pt_1Ag_{13}$ and $Pd_1Ag_{13}$ not only had open-shell electronic configurations, but also open geometric cores. By contrast, we found that the subtle difference between the kernel bond lengths in $Pt_1Ag_{13}$ and $Pd_1Ag_{13}$ was observed. The Pt-Ag length (on average, 2.747 Å) was slightly longer than the Pd-Ag length (on average, 2.742 Å), while the Ag-Ag lengths in the kernel of the $Pt_1Ag_{13}$ (on average, 2.886 Å) were slightly shorter than those of the $Pd_1Ag_{13}$ (on average, 2.892 Å) (Supplementary Fig. 15). Additionally, five Ag atoms located at the waist of the $M_{12}$ kernel were bounded to five $PPh_3$ ligands, whereas the either side six Ag atoms were connected to six S atoms from two $Ag(SR)_3(PPh_3)$ motifs. Only one type of $Ag_k$-S-$Ag_s$ mode was observed for $Pt_1Ag_{13}$ and $Pd_1Ag_{13}$ (Supplementary Fig. 16). The Ag-S lengths (on average, 2.550 Å) of $Pt_1Ag_{13}$ were almost identical to that of $Pd_1Ag_{13}$ (on average, 2.549 Å), while the Ag-P lengths (on average, 2.461 Å) of $Pt_1Ag_{13}$ was slightly shorter than that of $Pd_1Ag_{13}$ (on average, 2.469 Å), suggesting a slightly more compact shell structure of the former (Supplementary Fig. 15). Subtle differences in bond lengths may arise from the different behaviors of Pt and Pd nucleation.

### Dual emission of Pt₁Ag₁₃ and Pd₁Ag₁₃ nanoclusters

The ultraviolet-visible (UV-vis) absorption spectra of $Pt_1Ag_{13}$ and $Pd_1Ag_{13}$ are shown in Fig. 2a and Supplementary Fig. 17, respectively. Specifically, three prominent absorption peaks centered at 390, 470, and 600 nm are observed for $Pt_1Ag_{13}$. For $Pd_1Ag_{13}$, three significant peaks are observed at 420, 497, and 680 nm. This red shift of $Pd_1Ag_{13}$

compared with that of $Pt_1Ag_{13}$ in optical absorption may be attributed to their different electronic structures, which is consistent with the previous investigations on $[PtAg_{24}(SR)_{18}]^{2-}$ and $[PdAg_{24}(SR)_{18}]^{2-}$ (SR = 2,4-dichlorobenzenethiol) NCs[44]. The PL spectra revealed the presence of DE in both NCs, as shown in Fig. 2a and Supplementary Fig. 18. Specifically, $Pt_1Ag_{13}$ in 2-Me-THF solution exhibited one visible peak centered at 660 nm and one NIR peak centered at 825 nm, with photoluminescent quantum yield (PLQY) determined to be 1.49%. Similarly, the $Pd_1Ag_{13}$ in 2-Me-THF solution shows DE centered at 748 and 830 nm, respectively, with the PLQY measured to be 0.07%. The difference in the PLQY between the two NCs may be caused by different electron affinity of Pt or Pd (i.e., the capability to attracting electron)[45,46]. The stronger electron affinity of Pt may lead to enhanced charge transfer abilities, resulting in higher PLQY in $Pt_1Ag_{13}$.

The dynamics of the DE in $Pt_1Ag_{13}$ and $Pd_1Ag_{13}$ were investigated by time-correlated single-photon counting. The average lifetimes of $Pt_1Ag_{13}$ were calculated to be approximately 1.148 and 1.886 μs, respectively, according to the fitting results of the decays of PL I and PL II, indicating that PL I and PL II may exhibit phosphorescence (Fig. 2b, Supplementary Fig. 19 and Supplementary Table 4). In addition, the PL I and PL II intensities were simultaneously enhanced under an $N_2$ atmosphere and reduced under an $O_2$ atmosphere (Supplementary Figs. 20–22). Furthermore, the peak at 415 nm corresponding to the characteristic absorption of 1,3-diphenylisobenzofuran in solution containing $Pt_1Ag_{13}$ decreased rapidly, confirming the presence of singlet oxygen (Supplementary Fig. 23). These findings indicated that PL I and PL II of $Pt_1Ag_{13}$ exhibit phosphorescence. Similar results were observed for $Pd_1Ag_{13}$, with average lifetimes of approximately 1.034 μs and 1.509 μs for PL I and PL II respectively, both exhibiting

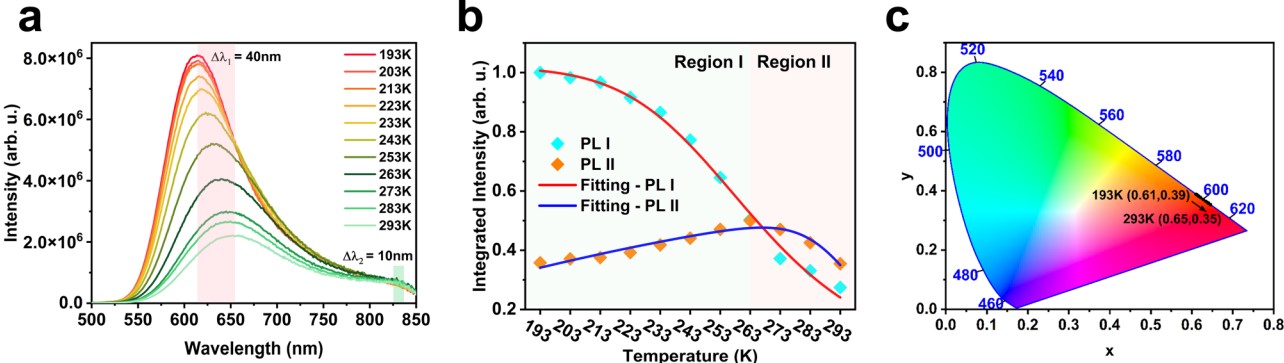

**Fig. 3 | Temperature-dependent PL spectra of Pt$_1$Ag$_{13}$. a** Variable-temperature PL spectra of Pt$_1$Ag$_{13}$ in 2-Me-THF. The color boxes represent the blue and red shift trends of PL I and PL II respectively. **b** Normalized integrated PL I and PL II intensities were fitted using Eqs. 1 and 2, respectively; the integration of PL II is separated as regions I and II. The colored box on the left represents Region I, and the colored box on the right represents Region II. **c** CIE 1931 color space chromaticity diagram showing the luminescence color change of Pt$_1$Ag$_{13}$ in the temperature range of 193-293 K.

phosphorescent characteristics. (Supplementary Figs. 24–28 and Supplementary Table 5).

We performed PL excitation (PLE) and wavelength-dependent PL analyzes to reveal the origin of DE in Pt$_1$Ag$_{13}$ and Pd$_1$Ag$_{13}$. The PLE spectra of Pt$_1$Ag$_{13}$ in 2-Me-THF for PL I and PL II were primarily located at 390, 425, 470, and 505 nm, with a new peak emerging at 600 nm upon PL II excitation (Fig. 2c). The maximum excitation peaks for PL I and PL II were different and located at 390 and 470 nm, respectively. The PLE spectra were not aligned with their absorption spectra, conversely to previously reported NCs, such as [Pt$_1$Ag$_{30}$(S-Adm)$_{14}$(Bdpm)$_4$Cl$_5$]$^{3+}$ (Bdpm = N, N-bis-(diphenylphosphino)methylamine) and Au$_2$Cu$_6$(S-Adm)$_6$(TPP)$_2$, where the PLE and absorption spectra were similar[31,47]. Hence, neither PL band was excited by the Pt$_1$Ag$_{11}$-core-based HOMO-LUMO transition.

Notably, the three-dimensional (3D) PL/PLE spectra showed a distinct difference between PL I and PL II of Pt$_1$Ag$_{13}$ (Fig. 2d). The PL I intensity exhibited dynamic fluctuations at different excitation wavelengths ranging from 310 to 630 nm, with the strongest emission observed at an excitation wavelength of 390 nm. Interestingly, no PL I emission was observed under 600 nm excitation. In contrast, PL II displayed a prominent peak emission under excitation at 470 nm, which diverged significantly from that of PL I in the 3D PL/PLE spectra (Fig. 2e and Supplementary Fig. 29). Furthermore, the excitation-dependent decay measurements of PL I and PL II were performed. Two exponential lifetimes were required to fit the PL I and PL II decays. The decay and rise of the $\tau_1$ and $\tau_2$ of PL I were accompanied by the rise and decay of the $\tau_1$ and $\tau_2$ of PL II (Fig. 2f, Supplementary Fig. 19, and Supplementary Table 4), which may result from the overlap of the two PL bands, similar to that reported for the Au$_{20}$ and Au$_{24}$ NCs[29,32]. Considering the correlation between the two PL bands, we deduced that PL I and PL II may originate from two distinct emitting states in Pt$_1$Ag$_{13}$. The PLE and wavelength-dependent PL spectra of Pd$_1$Ag$_{13}$ were also analyzed (Supplementary Figs. 30, 31). It was found that both PL I and PL II were present under excitation at 420, 497, and 600 nm. However, only PL II emission was observed under excitation at 680 nm. The result of excitation-dependent decay measurements of PL I and PL II in Pd$_1$Ag$_{13}$ was similar to that observed in Pt$_1$Ag$_{13}$ (Supplementary Table 5). Therefore, we posited that PL I and PL II in Pd$_1$Ag$_{13}$ may also stem from two distinct emitting states. Moreover, time-tracking UV-vis and PL spectra of Pt$_1$Ag$_{13}$ and Pd$_1$Ag$_{13}$ in 2-Me-THF solution were performed at room temperature, indicating its photo-stability for several hours (Supplementary Figs. 32, 33). Additionally, to determine whether the aggregates induced DE, PL tests on the Pt$_1$Ag$_{13}$ and Pd$_1$Ag$_{13}$ in 2-Me-THF solution at various concentrations were also conducted. As shown in Supplementary Figs. 34, 35, DE persisted even at low concentrations,

eliminating the possibility of DE due to the aggregation-induced emission effect.

## Temperature-dependent photoluminescence of Pt$_1$Ag$_{13}$ and Pd$_1$Ag$_{13}$ nanoclusters

To further understand the nonradiative relaxation process of the two emitting states, temperature-dependent steady-state PL measurements of Pt$_1$Ag$_{13}$ and Pd$_1$Ag$_{13}$ in 2-Me-THF were performed (Fig. 3a, Supplementary Fig. 36a). Upon decreasing the temperature from 293 to 193 K, the PL I and PL II peaks of Pt$_1$Ag$_{13}$ exhibited blue shifts of 40 and 10 nm, respectively. The visualized color change of Pt$_1$Ag$_{13}$ with temperature and the chromaticity coordinates $x$ and $y$ were plotted on a CIE 1931 color space chromaticity diagram, showing a shift from reddish-orange (CIE: 0.65, 0.35) to orange (CIE: 0.61, 0.39), as shown in Fig. 3c. Notably, the PL I intensity increased 3.64-fold, while the PL II intensity remained essentially unchanged in the temperature range of 293-193 K (Supplementary Table 6). Additionally, the temperature-dependent intensity for PL I and PL II of Pt$_1$Ag$_{13}$ were quantitatively depicted in Fig. 3b. The behavior of PL I band of Pt$_1$Ag$_{13}$ was more typical, with the initial intensity $I_O$ decreasing above 193 K owing to thermally activated quenching. With only one dominant nonradiative channel, this quenching process can be described by the Arrhenius expression[48]:

$$I(T) = \frac{I_0}{1 + ae^{\frac{-E_a}{k_B T}}} \tag{1}$$

In our model, we considered a single dominant phonon-assisted nonradiative channel, where "$a$" represents the ratio of nonradiative and radiative probabilities, and "$E_a$" denotes the activation energy of the quenching channel. When Eq. 1 to the temperature dependence of the PL I intensity was applied, we obtained $E_a$ values of 48.94 meV for Pt$_1$Ag$_{13}$.

The temperature dependence of the PL II emission of Pt$_1$Ag$_{13}$ exhibited more complex behavior. The intensity of the PL II peak of Pt$_1$Ag$_{13}$ increased with temperature and reached its maximum at approximately 263 K (region I). Thereafter, it started to decrease as the temperature increased (region II). In region II, the decrease in PL II intensity followed a trend similar to that of the PL I band, indicating a nonselective thermally activated nonradiative relaxation pathway (Fig. 3b). However, in region I, an additional non-radiative channel appeared. To account for this dual-quenching behavior, Arrhenius fitting can be adapted to incorporate a second quenching term,

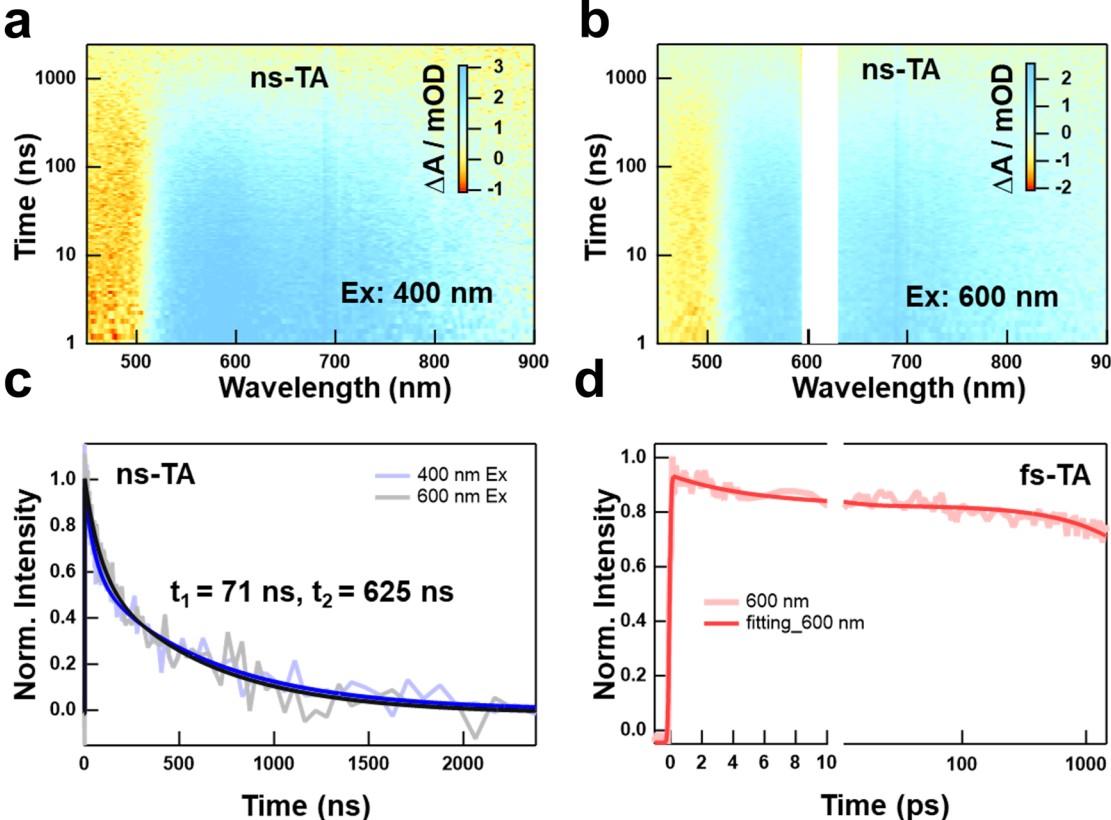

**Fig. 4 | Excited-state dynamics of Pt$_1$Ag$_{13}$. a, b** The ns-TA data of Pt$_1$Ag$_{13}$ under 400 nm (**a**) excitation and 600 nm (**b**) excitation, all time-resolved spectroscopy measurements are conducted with nitrogen protection. **c** The kinetic traces at 560 nm of Pt$_1$Ag$_{13}$ under 400 nm excitation and 600 nm excitation. The dark curves represent the fitting results of experimental data (light curves). **d** The kinetic trace at 600 nm extracted from the fs-TA data.

resulting in the following modified expression:

$$I(T) = \frac{I_0}{1 + a_1 e^{\frac{-E_{a_1}}{k_B T}} + a_2 e^{\frac{-E_{a_2}}{k_B T}}} \tag{2}$$

The two competing processes ($a_1 > 0$; $a_2 < 0$) combined to produce the maximum using the fitting parameters from Supplementary Table 7.

The $a$ value of Pt$_1$Ag$_{13}$ decreased drastically from 160.26 (PL I) to 39.56 and -8.54 (PL II $a_1$ and $a_2$), indicating less dependence on the surface motif vibration-induced nonradiative decay in PL II. Although the low-frequency phonon modes were typically attributed to the Au-Au vibrations of the metal kernel in Au NCs, it was challenging to isolate Pt$_1$Ag$_{13}$ as two nonradiative quenching channels in region I. The origin of the nonradiative quenching channel in region I for Pt$_1$Ag$_{13}$ may be related to the thermally acclerated internal conversion (IC) between the two emitting states. In other words, the increase in the PL II emission intensity in region I was accompanied by the decrease of radiative PL I transition.

The temperature-dependent steady-state PL behavior of Pd$_1$Ag$_{13}$ in 2-Me-THF closely resembled that observed in Pt$_1$Ag$_{13}$. The intensity of PL I was increased by 2.88 times, along with a 20 nm blue shift, while the intensity of PL II reached its maximum at 223 K with a 10 nm blue shift, as the temperature decreased. (Supplementary Fig. 36a, b and Supplementary Table 8). The CIE coordinates of (0.71, 0.29) were identical upon temperature decreasing (Supplementary Fig. 36c). The calculated $E_a$ values was 115.73 meV and the $a$ value of Pd$_1$Ag$_{13}$ also showed a sharp decrease from 115.73 (PL I) to 58.27 and -10.43 (PL II $a_1$ and $a_2$, Supplementary Table 9). Hence, we concluded that in both Pt$_1$Ag$_{13}$ and Pd$_1$Ag$_{13}$, PL I may originate from the core-shell CT

states, as it was more sensitive to low temperatures owing to the suppression of nonradiative relaxation processes. While PL II, which showed less dependence on temperature, may originate from core-based states.

To understand the photophysics of DE, we performed time-resolved transient absorption (TA) spectroscopy measurements with Pt$_1$Ag$_{13}$. We first looked into the nanosecond relaxation dynamics of Pt$_1$Ag$_{13}$ by performing ns-TA with excitation of 400 nm and 600 nm. Figure 4a showed the ns-TA data map with excitation of 400 nm that consisted of a negative band at 475 nm and a positive band across 520 nm to 900 nm. The negative band could be assigned to the ground state bleach (GSB) signal which coincided with the UV-vis absorption spectrum as shown in Fig. 4a, b, and the positive band was the excited state absorption (ESA) of the triplet state. The ns-TA data presented a monotonous decay without spectral shift, suggesting no new transient species were generated. The ns-TA data map (Fig. 4b) under 600 nm excitation was similar to that excited at 400 nm, which may be because the excited state dynamics of Pt$_1$Ag$_{13}$ under the different-energy excited laser were very close to each other thus making the ns-TA set-up cannot distinguish the differences. This was further demonstrated by the almost overlapped kinetic traces at 560 nm with an average lifetime of less than 1 μs ($t_1 = 71$ ns, $t_2 = 625$ ns, Fig. 4c), which was close to the lifetime obtained from the fluorescence lifetime (around 1 μs). We also conducted the fs-TA measurements under 400 nm excitation, the kinetic traces at 600 nm with a lifetime larger than 2 ns were displayed in Fig. 4d, and no more new transient components were obtained. These results indicated that TA spectroscopy mainly probed the dynamics of core-shell CT excited state (PL I), which was much stronger than core-based one (PL II). These results were consistent with the ns-TA test results of reported Au$_{20}$[32].

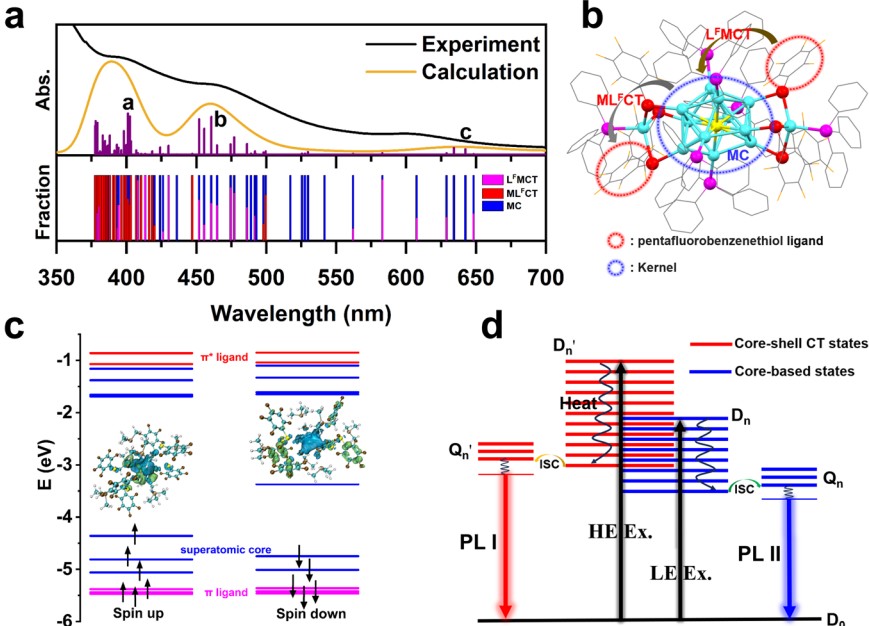

**Fig. 5 | DFT Calculations of Pt₁Ag₁₃. a** Experimental and calculated absorption spectrum of Pt₁Ag₁₃ with contributions from metal-centered transition (MC), metal-ligand charge transfer (ML$^F$CT) and ligand-metal charge transfer (L$^F$MCT) excited states. **b** Intuitive diagram of charge transfer excited states. **c** Molecular orbital scheme of Pt₁Ag₁₃ showing the energy levels of frontier orbitals. (superatomic orbitals in Ag core (red), ligand-based π orbitals (blue), and ligand-based π* orbitals (green), and the transition densities of the dominant MC (left) and MLCT (right)

transitions are also depicted (hole: blue, electron: green).) **d** Proposed schematic dual emission mechanism of Pt₁Ag₁₃. (HE high energy, LE low energy, Ex excitation, ISC intersystem crossing, $D_0$ doublet ground state, $D_n$ core-based doublet excited states, $D_n'$ core-shell CT doublet excited states, $Q_n$ core-based quartet excited states, $Q_n'$ core-shell CT quartet excited states, PL I and PL II phosphorescence emission processes, respectively).

## Theoretical calculations on electronic structures and excited states

To further investigate the nature of the dual-emitting states, time-dependent density functional theory (TD-DFT) calculations were performed on the optimized structure of the Pt₁Ag₁₃ NC. The Pt₁Ag₁₃ core is an open-shell superatom with seven superatomic valence electrons (7e). The concept of "open-shell" superatom was previously applied to Ag₃₉ and Ag₃₀₇[49,50], corresponding to open-shell 17-electron and 135-electron superatoms, respectively. Supplementary Fig. 37 illustrates the highest occupied molecular orbitals (α-HOMOs and β-HOMOs) and the lowest unoccupied molecular orbitals (α-LUMOs and β-LUMOs), which are confined to the metal kernel and exhibit a typical superatomic shell ($S^2P^5$). The details of the superatomic shell of Pt₁Ag₁₃ are presented in Supplementary Table 10.

The calculated UV-vis absorption spectrum of Pt₁Ag₁₃ in 2-Me-THF agrees well with the experimental results (Fig. 5a). The absorption spectrum of Pt₁Ag₁₃ can be divided into three regions, and three states (a, b and c) with higher oscillation intensities specifically chosen from the numerous excitation states. The first region locates at $\lambda < 425$ nm (Peak a, $\lambda_{max} = 401$ nm), the second is at 425 nm $< \lambda < 550$ nm (Peak b, $\lambda_{max} = 460$ nm) and the third is at $\lambda > 550$ nm (Peak c, $\lambda_{max} = 634$ nm). Contributions from three types of transitions in absorption spectrum, including metal-centered transition (MC), metal-ligand charge transfer (ML$^F$CT) and ligand-metal charge transfer (L$^F$MCT) excited states, are also investigated and revealed in Fig. 5a, b. More details of the frontier orbitals, excited states and contributions of metal and ligand fragments are given in Fig. 5c and Supplementary Tables 10, 11.

The low-energy (LE) absorption above 550 nm is dominated of MC states, where the transitions occur within the frontier superatomic orbitals, eg. from occupied super $P_{x,y,z}$ to unoccupied super $D_{xy,yz,zx,z2}$ orbitals. The intermediate range of the spectrum (from 425 nm to 550 nm) involves the mixed excited states that are combined with ML$^F$CT, L$^F$MCT and MC, where contributions of PFBT ligands are involved and TPP ligands are neglectable in these transitions. The high-

energy (HE) states below 425 nm could be primarily ascribed to ML$^F$CT states, which involves the transitions from superatomic orbitals of metal core to π* orbitals in PFBT ligands. As electron-poor character of the 7e superatomic Pt₁Ag₁₃ core ($S^2P^5$, one less valence electron from the 8e shell-closure), MLCT emission might hardly occur. However, PFBT ligands, with fluorine substituted benzene groups, show intense electronegativity and give rise to the low-lying the π* orbitals that is more easily accessible, where the transitions from metal core to PFBT ligands are observed. Therefore, ligand-effect of PFBT play an important role in this series of transition states. In short, high-, mid-, and low-energy transitions are denoted as core-shell CT states, mixed states and core-based states, respectively.

Excitation of Pt₁Ag₁₃ into the HE absorption at 390 nm results in two emission bands (PL I and PL II), corresponding to core-shell CT and core-based absorptions, while excitation into the LE absorption band at 600 nm results in the emission mirroring the core-based absorption bands (PL II) (see Fig. 2d, e). The ratio of the core-shell CT states consistently decreases with wavelengths ranging from 390 to 630 nm (see Fig. 5a), which aligns with the evolutionary trend of the PL I intensity observed in Supplementary Fig. 38. This confirms that the CT contribution of phosphorescence in PL I primarily occurs from the core-shell CT states. Similarly, the evolutionary trends of PL II and core-based proportions were in agreement. Hence, it is evident that the PL I and PL II emissions originate from two distinct emitting states: core-shell CT and core-based states, respectively. Previous studies[51] give evidence for the dual emission coming from two different emissive states in a single complex and thus violating Kasha's rule[52].

TD-DFT calculations on the optimized structure of Pd₁Ag₁₃ was also performed, show similar nature of electron transitions with Pt₁Ag₁₃. The Pd₁Ag₁₃ also has a 7e open-shell superatomic core. The calculated UV-vis absorption spectrum of Pd₁Ag₁₃ in 2-Me-THF agrees well with the experimental results (Supplementary Fig. 39). The high- (419 nm), mid- (493 nm), and low- (640 nm) energy transitions are also

classified as core-shell CT states, mixed states and core-based states respectively according to Supplementary Tables 12, 13.

Based on the above experimental and theoretical results, the proposed DE mechanism of open-shell $Pt_1Ag_{13}$ is given in Fig. 5d. The low-lying doublet and quartet states are classified into core-based states ($D_1$, $Q_1$) and core-shell CT states ($D_1'$, $Q_1'$), respectively, and the details can be found in Supplementary Tables 11, 14. The HE absorption is primarily attributed to the core-shell CT, while the LE absorption is attributed to the inner superatomic core. These two types of electronic states experience rapid relaxation from their higher states to the lowest core-based state ($D_1$) and the lowest core-shell CT state ($D_1'$), respectively. After that, they undergo ISC processes to the core-based $Q_1$ and core-shell CT $Q_1'$ states ($D_1 \rightarrow Q_1$, $D_1' \rightarrow Q_1'$) due to the intense spin-orbit coupling (SOC) interactions induced by Pt and Ag atoms and their close energy levels. As a result, a visible PL I emission is observed from the core-shell CT states, and an NIR PL II emission is observed from the core-based states. As the NIR PL II emission originates from the core states, it is found to be less affected by temperature variation. Given the analogous electron transition characteristics between $Pd_1Ag_{13}$ and $Pt_1Ag_{13}$, the proposed mechanism can also be applied to $Pd_1Ag_{13}$. This DE character of $Pt_1Ag_{13}$ and $Pd_1Ag_{13}$ NC largely depends on the role of electronegative PFBT ligands, providing an effective blueprint for designing materials with DE.

## Discussion

Atomically precise $Pt_1Ag_{13}(PFBT)_6(TPP)_7$ and $Pd_1Ag_{13}(PFBT)_6(TPP)_7$ NCs with 7-electron open-shell configurations were synthesized using a metal-controlled one-pot methodology. Both NCs showed dual-quartet phosphorescent emission in visible and NIR regions, with PLQYs of 1.49% and 0.07%, respectively. This unusual dual-quartet phosphorescent emission may be attributed to two-family states, that is, core-shell CT and core-based states. Moreover, ISC occurs from the lowest-lying doublet state to the corresponding quartet state induced by the small energy gap and heavy-atom effect via SOC. Hence, phosphorescence was observed. Specifically, the high-energy visible emission and the low-energy NIR emission were ascribed to the core-shell CT and core-based states, respectively. Moreover, the presence of PFBT ligands is crucial for the DE in the two NCs, as their low-lying $\pi^*$ levels facilitate energetically feasible core-shell transitions. This work reports the dual-quartet phosphorescent emission in an open-shell NC, which will enrich our understanding of the DE mechanism and may shed new light on the applications of NCs in magnetoluminescence and novel optoelectronic devices.

## Methods

### Materials and reagents

Potassium tetrachloroplatinate (II) ($K_2PtCl_4$, 99%), Palladium(II) chloride ($PdCl_2$, 99%), Silver nitrate ($AgNO_3$), pentafluorobenzenethiol (PFBT), sodium borohydride ($NaBH_4$), triphenylphosphine (TPP), and 1,3-diphenylisobenzofuran (DPBF) were purchased from Shanghai Macklin Biochemical Co., Ltd. Solvents, including dichloromethane (DCM, HPLC grade), methanol (MeOH, HPLC grade), and n-hexane ($n$-Hex, HPLC grade) were purchased from Shanghai Aladdin Bio-Chem Technology Co., Ltd. The ultrapure water ($\geq$18.2 M$\Omega$) used in this work was purified on a Millipore system (Millipore).

### Synthesis of the $Pt_1Ag_{13}(PFBT)_6(TPP)_7$ nanocluster

40 mg $AgNO_3$ and 10 mg of $K_2PtCl_4$ were dissolved into 10 mL methanol in a 50 mL round-bottomed flask. The solution was stirred vigorously at room temperature for 10 min. The solution immediately turns brown. Subsequently, 40 μL PFBT was added into the flask. After 5 min of reaction, 200 mg $PPh_3$ dissolved in 10 mL $CH_2Cl_2$ was added under vigorous stirring. The color of the solution becomes transparent. Then, 2 mL of an aqueous solution of $NaBH_4$ (20 mg/mL) was added quickly to the reaction mixture under vigorous stirring. The

solution color immediately changed from brown to black. The reaction was subsequently carried out for a duration of 12 hours under a $N_2$ atmosphere at room temperature, shielded from light, until the solution transitioned into a distinctive orange hue. It indicated that clusters are formed. The aqueous phase was then removed. The organic phase was washed several times with water and methanol. The crystals were crystallized from $CH_2Cl_2$/hexane at room temperature in the dark and afford orange rhombohedral single crystals after 5 days. The yield was 11.5% based on the Ag element (calculated from the $AgNO_3$) for synthesizing the $Pt_1Ag_{13}$ nanocluster.

### Synthesis of the $Pd_1Ag_{13}(PFBT)_6(TPP)_7$ nanocluster

The 10 mg of $K_2PtCl_4$ and 40 mg of $AgNO_3$ reagents used to synthesize $Pt_1Ag_{13}(PFBT)_6(TPP)_7$ were substituted by 12 mg of $PdCl_2$ and 40 mg of $AgNO_3$, respectively. Other conditions remained unchanged, and the $Pd_1Ag_{13}(PFBT)_6(TPP)_7$ nanocluster was obtained. The yield was 10.3% based on the Ag element (calculated from the $AgNO_3$) for synthesizing the $Pd_1Ag_{13}$ nanocluster.

### Detection of singlet oxygen ($^1O_2$) generation

DPBF was used to detect the $^1O_2$ generation by $Pt_1Ag_{13}$ or $Pd_1Ag_{13}$. In brief, DPBF in ethanol was prepared, to which the 2-Me-THF solution of $Pt_1Ag_{13}$ or $Pd_1Ag_{13}$ was added to give final concentrations of 1 mM for DPBF and $3.5 \times 10^{-2}$ μM for $Pt_1Ag_{13}$ or $Pd_1Ag_{13}$. The mixed solution was irradiated by Xe lamp, and the adsorption spectra were recorded on an Agilent 8453 UV-vis spectrophotometer.

### Characterizations

Electrospray ionization mass spectrometry (ESI-MS) measurements were performed by Waters XEVO G2-XS QTof mass spectrometer. Thermogravimetric analysis (TGA) was carried out using a thermogravimetric analyzer (DTG-60H, Shimadzu Instruments, Inc.). X-ray photoelectron spectroscopy (XPS) measurements were performed using a Thermo ESCALAB 250 configured with a monochromated Al Kα (1486.8 eV) 150 W X-ray source, 0.5 mm circular spot size, a flood gun to counter charging effects, and an analysis chamber base pressure lower than $1 \times 10^{-9}$ mbar, and the data were collected with FAT of 20 eV. Inductively coupled plasma-atomic emission spectrometry (ICP-AES) measurements were performed on an Atomscan advantage instrument from Thermo Jarrell Ash Corporation (USA). Energy-dispersive X-ray spectroscopy (EDS) analyzes were performed on a JEOL JEM-2100F FEG TEM operated at 200 kV. Electron paramagnetic resonance (EPR) measurement was conducted on a Bruker X-band (9.4 GHz) EMS plus 10/12 spectrometer. All UV-vis spectra of the nanoclusters were recorded using an Agilent 8453. Photoluminescence (PL) spectra were measured using a HORIBA FluoroMax+ spectrofluorometer. Absolute PL quantum yields (PLQYs) and emission lifetimes were measured with dilute solutions of nanoclusters on a HORIBA FluoroMax-4P. The data collection for single-crystal X-ray diffraction (SC-XRD) of all nanocluster crystal samples was carried out on Stoe Stadivari diffractometer under nitrogen flow, using graphite-monochromatized Cu Kα radiation ($\lambda$ = 1.54186 Å). Data reductions and absorption corrections were performed using the SAINT and SADABS programs, respectively. Fs-TA measurements were performed on a home-built setup and ns-TA measurement were performed using a commercial spectrometer (Nano100, Time-Tech Spectra)[53].

### Theoretical calculations

The structures of liganded $Pt_1Ag_{13}$ nanocluster was fully optimized by using density functional theory (DFT) method at B3LYP/def2SVP[54,55] level of theory with Grimme D3 corrections[56], and verified to be true minima by frequency check (Supplementary Data 1). The benzene groups in TPP ligands in experimental structure are replaced by methyl groups to simplify the structure, which have little influence on its electronic characters. Calculated UV absorption spectrum is obtained

by time-dependent density functional theory (TD-DFT)[57,58] calculation. Benchmark for different functionals, including PBE[59], hybrid functional with different HF compositions (B3LYP, PBE0[60], M06-2X[61]), and range-separated functional with different $\alpha/\beta/\omega$ parameters (LC-BLYP[62,63], CAM-B3LYP[64], $\omega$B97XD[65]), are carried out for TD-DFT calculation by comparing with the experimental data. The $\omega$ parameters in LC-BLYP are optimized to be 0.01 by using the optDFT$\omega$ package proposed by Lu Tian[66]. Among these, the result of B3LYP functional is most comparable to the experimental spectra (Supplementary Fig. 40). Therefore, B3LYP functional is finally chosen in our work. Compositions of molecular orbitals are analyzed based on natural atomic orbital (NAO) partition[67]. All calculations are carried out in Gaussian 16[68] and Multiwfn[69] package, and the Kohn-Sham orbitals are visualized in the Visual Molecular Dynamics (VMD) program[70].

## Data availability

Data supporting the findings of this work are available within the article and its Supplementary Information. The data that support the findings of this study are available from the corresponding author upon request. The X-ray crystallographic coordinates for the structures reported in this article have been deposited at the Cambridge Crystallographic Data Centre (CCDC) under deposition number CCDC $Pt_1Ag_{13}$ (2300261), and $Pd_1Ag_{13}$ (2304388). These data can be obtained free of charge from the CCDC via www.ccdc.cam.ac.uk/data_request/cif.

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

## Acknowledgements

We acknowledge the financial support provided by the National Natural Science Foundation of China 22371003 (M. Z. Z.), 21871001 (M. Z. Z.), 22103001 (C. X.) and 21901001 (S. J.), Natural Science Fund of the Education Department of Anhui Province 2022AH040018 (S. J.) and 2023AH050108 (D. Q. H.). The authors also appreciate the esteemed Rodolphe Antoine, an expert from the University of Lyon in France, for his insightful explanation of the luminescence mechanism.

## Author contributions

C.F. carried out the experiments, analyzed the data and wrote the manuscript. C.X. and L.X.Q. completed DFT calculations and assisted in the mechanism discussion. W.Z. and M.Z. participated in the Fs and ns-TA spectroscopy measurements. D.T. assisted in the synthesis and measurements. D.Q.H. and M.Z.Z. designed the project, analyzed the data, and revised the manuscript. S.J. assisted in the X-ray structure analysis.

## Competing interests

The authors declare no competing interests.
