## [Peer Review File · Nature Communications]

Dual-quartet phosphorescent emission in the open-shell M1Ag13 (M = Pt, Pd) nanoclustersReviewer #1 (Remarks to the Author):

After a careful evaluation of the manuscript entitled "Dual-quartet phosphorescent emission in the open-shell Pt1Ag13 nanocluster," I cannot recommend its publication in its current form. However, after addressing the following remarks within a new submission, I believe it could be suitable for Nature Communications.

1) Firstly, from a theoretical chemistry standpoint, the weight and criticism of how the theoretical calculations were conducted do not match the level of detail provided for the experimental part. Consequently, I cannot fully accept the equal contributions of the first two authors of the manuscript. The TD-DFT calculations were portrayed as simple and straightforward, which they are not. There is a significant lack of information regarding these calculations, with only few details available in one of the supplementary information (SI) files. Readers have no insights into the chosen level of theory until a thorough examination of the SI files. However, such crucial information should be included, at least, in the captions of Figures 1 and 4, for instance.

2) Secondly, essential tests regarding the level of theory used particularly for the TD-DFT, which are mandatory, were not performed. For instance, given that the manuscript is centered on the novel synthesis of Pt1Ag13 and Pd1A13 nanoclusters and their phosphorescent emission, it is essential to conduct basis-set optimization tests alongside functional selection and comparison with existing literature. Testing various combinations of functionals and basis sets, including range-separated functionals, e.g., LC-BLYP, (<https://www.nature.com/articles/s41557-023-01137-w>) can offer valuable insights into accurately capturing the relevant electronic transitions. The chosen level of theory, namely B3LYP/def2SVP, for the TD-DFT calculations, lacks proper justification and precision for this complex scenario. Therefore, conducting additional tests or providing a rationale reason for the chosen level of theory is mandatory to guarantee a more accurate description of electronic states and related transitions.

3) The authors claim that the mechanism behind DE is not demonstrated with certainty (also mentioned in Line 45 of the Introduction section). They need to improve and develop better those explanations/statements, supported by grounded experimental evidence and properly applied theoretical calculations references. Thus, the phosphorescent emission itself is explained by the authors as core-shell CT and core-core Ts. Therefore, I partially agree with the novelty regarding synthesis and experimental efforts, but not with the subsequent statement in lines 300-303 regarding DE phospho-mechanism, "which will enrich our understanding of the DE mechanism."

Concluding, with the necessary revisions outlined above particularly on the Theoretical Calculations, a resubmission could potentially make the manuscript suitable for publication in Nature Communications.

Reviewer #2 (Remarks to the Author):

Zhu and his co-workers reported the preparation of two open-shell M1Ag13 alloy nanoclusters (NCs) (M = Pt and Pd) using a mixed-ligand synthetic method. While the Pt1Ag13 NC exhibits interesting phosphorescent dual emission, there is no evidence that the comparable Pd1Ag13 NC behaves similarly in terms of photoluminescence (PL). In Supplementary Figs. 18 and 24, the second emission peak of the Pd1Ag13 NC around 830 nm is exceedingly faint and difficult to detect. Surprisingly, the authors did not provide any lifetime data, excitation spectra, or temperature-dependent PL analysis to support the claim that the Pd1Ag13 NC has dual emission, nor did they look into the connection between the alloy metal cores and the PL properties of these two structurally highly relevant NCs. Due to the omission of thorough PL characterization for the Pd1Ag13 NC and a lack of mechanistic insights to illustrate the PL differences between these two open-shell alloy NCs, this reviewer concludes that the quality of the current manuscript does not meet the standards of Nature Communications.

In addition to the serious issues mentioned above, the authors should take the following into consideration:

1. Generally speaking, 7-electron open-shell NCs have poor stability. The authors must offer evidence to show that the two NCs remain unchanged during the PL measurements in 2-methyltetrahydrofuran. The exploration of dual emission in other organic solvents is highly recommended.
2. It is insufficient to investigate the origin of dual emission just using DFT calculations. As the most important part of the current work, the authors should carry out additional experimental studies to obtain comprehensive information regarding the photoexcited species of the M1Ag13 NCs. Transient absorption measurements along with the corresponding analysis are necessary (see Nat. Commun. 2020, 11, 2897; J. Am. Chem. Soc. 2022, 144, 19243; Sci. Adv. 2023, 9, eadg3587, etc.)
3. A highly relevant reference on dual-emission mechanisms is missing (see J. Phys. Chem. Lett. 2021, 12, 1514).

Reviewer #3 (Remarks to the Author):

Although the dual-emission (DE) of metal nanoclusters has been widely studied due to its promising applications in ratio metric sensing, bioimaging, novel optoelectronic devices, etc., there are rare examples of DE with open-shell configurations. Moreover, the lack of fundamental understanding between structure and DE properties in open-shell NCs with doublet or quartet emissions impedes exploring the controllable synthesis of NCs with DE. This work reports the synthesis and photophysical properties of dual-emissive nanoclusters M1Ag13(PFBT)6(TPP)7 (M1Ag13; M = Pt, Pd) with open shell electronic structure, and clarifies the DE mechanism attributed to quartet states which involves core-shell CT and core-based excitation, based on time-resolved spectroscopy and theoretical calculations. This paper reports a novel point about dual-quartet phosphorescent emission in the open-shell Pt1Ag13 nanocluster. Thus, I strongly recommend publishing this interesting work in Nature Communications.

Minor revisions:

1. A closed-shell electronic configuration is always preferred for clusters of group 11 elements, and superatom theory and electron-counting rules have been developed to explain cluster stability. So how is the stability of Pt1Ag13 and Pd1Ag13 NCs with one unpaired electron? If stable, how can they be stabilized?
2. The authors mention that PL I is centered at 650 nm while PL II is centered at 825 nm in PtAg13, and further assign PL I to the core-shell charge transfer state and PL II to the core state. My question is how can the author make sure the DE might not come from the cluster itself but from some impurity or decompositions? To answer this question, it would be useful to show the ¹H NMR spectra of PtAg13.
3. The authors confirm the singlet oxygen production by decreasing the characteristic absorption peak of 1,3-diphenylisobenzofuran, indicating that PtAg13 is phosphorescent. The singlet oxygen experiment in PdAg13 should also be supplemented to identify the PL properties.
4. It seems the authors take some simplifications for ligands in theoretical calculation. Does it have any influence on PL characters?
5. It is recommended to explain the chromaticity coordinates x and y in Fig. 3C and to describe how to make this plot.
6. The yields of Pt1Ag13 and Pd1Ag13 NCs were not given in the SI.
7. The format of several references is incorrect (i.e., the capital letter, missing subscript, Italic, etc.).

We thank all reviewers for their helpful comments. The point-by-point responses are shown in blue in this letter, and revisions are in red.

Reviewer #1 (Remarks to the Author):

After a careful evaluation of the manuscript entitled "Dual-quartet phosphorescent emission in the open-shell $\text{Pt}_1\text{Ag}_{13}$ nanocluster," I cannot recommend its publication in its current form. However, after addressing the following remarks within a new submission, I believe it could be suitable for Nature Communications.

1) Firstly, from a theoretical chemistry standpoint, the weight and criticism of how the theoretical calculations were conducted do not match the level of detail provided for the experimental part. Consequently, I cannot fully accept the equal contributions of the first two authors of the manuscript. The TD-DFT calculations were portrayed as simple and straightforward, which they are not. There is a significant lack of information regarding these calculations, with only few details available in one of the supplementary information (SI) files. Readers have no insights into the chosen level of theory until a thorough examination of the SI files. However, such crucial information should be included, at least, in the captions of Figures 1 and 4, for instance.

Response: Many thanks for your valuable comments.

In the revised manuscript, DE mechanism of $\text{Pt}_1\text{Ag}_{13}$ NC was further clarified by investigating the detail contributions from metal-center (MC), metal-ligand charge transfer (ML^{FCT} , L^{F} represent the PFBT ligands) and ligand-metal charge transfer (L^{FMCT}) in excited states, and the origin of the two emissive peaks was revealed accordingly. Moreover, DE mechanism of $\text{Pd}_1\text{Ag}_{13}$ NC was also investigated using TD-DFT calculation, which was similar to the $\text{Pt}_1\text{Ag}_{13}$. Relative discussion has been improved in the revised manuscript, and Fig. 4 (Fig. 5 in the revised version) has been rearranged.

Revised manuscript: "DFT Calculations on Electronic Structures and Excited States"

To further investigate the nature of the dual-emitting states, time-dependent density functional theory (TD-DFT) calculations were performed on the optimized structure of the $\text{Pt}_1\text{Ag}_{13}$ NC. The $\text{Pt}_1\text{Ag}_{13}$ core is an open-shell superatom with seven superatomic valence electrons (7e). The concept of "open-shell" superatom was previously applied to Ag_{39} and

Ag₃₀₇^{50,51}, corresponding to open-shell 17-electron and 135-electron superatoms, respectively.

Supplementary Fig. 37 illustrates the highest occupied molecular orbitals (α -HOMOs and β -HOMOs) and the lowest unoccupied molecular orbitals (α -LUMOs and β -LUMOs), which are confined to the metal kernel and exhibit a typical superatomic shell (S^2P^5). The details of the superatomic shell of Pt₁Ag₁₃ are presented in Supplementary Table 10.

New Supplementary Figure 37. Superatomic shell of Pt₁Ag₁₃ core in its highest occupied molecular orbitals (HOMOs) and lowest unoccupied molecular orbitals (LUMOs).

Fig. 5: DFT Calculations of Pt₁Ag₁₃.

a Left: experimental and calculated UV-vis spectrum of Pt₁Ag₁₃ with contributions from metal-centered transition (MC), metal-ligand charge transfer (ML^FCT) and ligand-metal charge transfer (L^FMCT) excited states. Right: charge transfer excited states intuitive diagram. **b** Molecular orbital scheme of Pt₁Ag₁₃ showing the energy levels of frontier orbitals (superatomic orbitals in Ag core (red), ligand-based π orbitals (blue), ligand-based π^* orbitals (green), and the transition densities of the dominant MC (left) and MLCT (right) transitions are also depicted (hole: blue, electron: green).) **c** Proposed schematic DE mechanism of Pt₁Ag₁₃. (HE: high energy, LE: low energy, Ex: excitation, ISC: intersystem crossing, D₀: doublet ground state, D_n: core-based doublet excited states, D_n' : core-shell CT doublet excited states, Q_n: core-based quartet excited states, Q_n' : core-shell CT quartet excited states, PL I and PL II: phosphorescence emission processes, respectively.)

The **calculated** UV-vis absorption spectrum of Pt₁Ag₁₃ in 2-Me-THF agrees well with the experimental results (Fig. 5a). The absorption spectrum of Pt₁Ag₁₃ can be divided into three regions, and three states (**a**, **b** and **c**) with higher oscillation intensities were specifically chosen

from the numerous excitation states. The first region locates at $\lambda < 425$ nm (Peak **a**, $\lambda_{\text{max}} = 401$ nm, blue in Fig. 5a), the second is at $425 \text{ nm} < \lambda < 550$ nm (Peak **b**, $\lambda_{\text{max}} = 460$ nm, green in Fig. 5a) and the third is at $\lambda > 550$ nm (Peak **c**, $\lambda_{\text{max}} = 634$ nm, orange in Fig. 5a). Contributions from three types of transitions in absorption spectrum, including metal-centered transition (MC), metal-ligand charge transfer (ML^{FCT}) and ligand-metal charge transfer (L^{FMCT}) excited states, are also investigated and revealed in Fig. 5a. More details of the frontier orbitals, excited states and contributions of metal and ligand fragments are given in Fig. 5b and Supplementary Tables 10, 11.

The low-energy (LE) absorption above 550 nm is dominated of MC states, where the transitions occur within the frontier superatomic orbitals, eg. from occupied super $P_{x,y,z}$ to unoccupied super D_{xy,yz,zx,z^2} orbitals. The intermediate range of the spectrum (from 425 nm to 550 nm) involves the mixed excited states that combined with ML^{FCT} , L^{FMCT} and MC, where contributions of PFBT ligands are involved and TPP ligands are neglectable in these transitions. The high-energy (HE) states below 425 nm could be primarily ascribed to ML^{FCT} states, which involves the transitions from superatomic orbitals of metal core to π^* orbitals in PFBT ligands. As electron-poor character of the 7e superatomic $\text{Pt}_1\text{Ag}_{13}$ core (S^2P^5 , one less valence electron from the 8e shell-closure), MLCT emission might hardly occur. However, PFBT ligands, with fluorine substituted benzene groups, show intense electronegativity and give rise to the low-lying the π^* orbitals that is more easily accessible, where the transitions from metal core to PFBT ligands are observed. Therefore, ligand-effect of PFBT play an important role in this series of transition states. In short, high-, mid-, and low- energy transitions are denoted as core-shell CT states, mixed states and core-based states, respectively.

Excitation of $\text{Pt}_1\text{Ag}_{13}$ into the HE absorption at 390 nm results in two emission bands (PL I and PL II), corresponding to core-shell CT and core-based absorptions, while excitation into the LE absorption band at 600 nm results in the emission mirroring the core-based absorption bands (PL II) (see Fig. 2d, e). The ratio of the core-shell CT states consistently decreases with wavelengths ranging from 390 to 630 nm (see Fig. 5a), which aligns with the evolutionary trend of the PL I intensity observed in Supplementary Fig. 38. This confirms that the CT contribution of phosphorescence in PL I primarily occurs from the core-shell CT states. Similarly, the evolutionary trends of PL II and core-based proportions were in agreement. Hence, it is evident

that the PL I and PL II emissions originate from two distinct emitting states: core-shell CT and core-based states, respectively. Previous studies⁵² give evidence for the dual emission coming from two different emissive states in a single complex and thus violating Kasha's rule⁵³.

TD-DFT calculations on the optimized structure of **Pd₁Ag₁₃** was also performed, show similar nature of electron transitions with **Pt₁Ag₁₃**. The **Pd₁Ag₁₃** also has a 7e open-shell superatomic core. The calculated UV-vis absorption spectrum of **Pd₁Ag₁₃** in 2-Me-THF agrees well with the experimental results (Supplementary Fig. 39). The high- (419 nm), mid- (493 nm), and low- (640) energy transitions are also classified as core-shell CT states, mixed states and core-based states respectively according to Supplementary Tables 12, 13.

Based on the above experimental and theoretical results, the proposed DE mechanism of open-shell **Pt₁Ag₁₃** is given in Fig. 5c. The low-lying doublet and quartet states are classified into core-based states (D_1 , Q_1) and core-shell CT states (D_1' , Q_1'), respectively, and the details can be found in Supplementary Tables 11, 14. The HE absorption is primarily attributed to the core-shell CT, while the LE absorption is attributed to the inner superatomic core. These two types of electronic states experience rapid relaxation from their higher states to the lowest core-based state (D_1) and the lowest core-shell CT state (D_1'), respectively. After that, they undergo ISC processes to the core-based Q_1 and core-shell CT Q_1' states ($D_1 \rightarrow Q_1$, $D_1' \rightarrow Q_1'$) due to the intense spin-orbit coupling (SOC) interactions induced by Pt and Ag atoms and their close energy levels. As a result, a visible PL I emission is observed from the core-shell CT states, and an NIR PL II emission is observed from the core-based states. As the NIR PL II emission originates from the core states, it is found to be less affected by temperature variation. Given the analogous electron transition characteristics between **Pd₁Ag₁₃** and **Pt₁Ag₁₃**, the proposed mechanism can also be applied to **Pd₁Ag₁₃**. This DE character of **Pt₁Ag₁₃** and **Pd₁Ag₁₃** NC largely depends on the role of electronegative PFBT ligands, providing an effective blueprint for designing materials with dual emissions.”

(Pages 14-18, lines 269-342)

New References:

52. Steube, J. et al. Janus-type emission from a cyclometalated iron(III) complex. *Nat. Chem.* **15**, 468-474 (2023).

53. Kasha, M. Characterization of electronic transitions in complex molecules. *Discuss. Faraday Soc.* **9**, 14-19 (1950).

2) Secondly, essential tests regarding the level of theory used particularly for the TD-DFT, which are mandatory, were not performed. For instance, given that the manuscript is centered on the novel synthesis of Pt₁Ag₁₃ and Pd₁Ag₁₃ nanoclusters and their phosphorescent emission, it is essential to conduct basis-set optimization tests alongside functional selection and comparison with existing literature. Testing various combinations of functionals and basis sets, including range-separated functionals, e.g., LC-BLYP, (<https://www.nature.com/articles/s41557-023-01137-w>) can offer valuable insights into accurately capturing the relevant electronic transitions. The chosen level of theory, namely B3LYP/def2SVP, for the TD-DFT calculations, lacks proper justification and precision for this complex scenario. Therefore, conducting additional tests or providing a rationale reason for the chosen level of theory is mandatory to guarantee a more accurate description of electronic states and related transitions.

Response: Many thanks for this valuable suggestion. Benchmark of functionals and basis-sets are performed in the revised version. Considering existing literatures of photoluminescence for liganded gold/silver clusters, we test series of functionals at def2SVP level in TD-DFT calculation for UV-vis absorption spectrum of Pt₁Ag₁₃, and the results are taken into comparison with experimental data. The functionals in our test include: **PBE**, hybrid functional with different HF compositions (**B3LYP**, **PBE0**, **M06-2X**), and range-separated functional with different $\alpha/\beta/\omega$ parameters (**LC-BLYP**, **CAM-B3LYP**, **ω B97XD**), where the ω parameters in LC-BLYP are optimized to be 0.01 by using the **optDFT ω** package proposed by Lu. The results in **New Supplementary Figure 40** reveal that the UV-vis spectrum obtained using B3LYP functional is most comparable to the experimental values. Details of DFT Calculations are revised in the Methods:

Revised manuscript: “DFT Calculations.

The structures of liganded Pt₁Ag₁₃ nanocluster was fully optimized by using density functional theory (DFT) method at B3LYP/def2SVP^{55,56} level of theory with Grimme D3 corrections⁵⁷, and verified to be true minima by frequency check. The benzene groups in TPP

ligands in experimental structure are replaced by methyl groups to simplify the structure, which have little influence on its electronic characters. Calculated UV absorption spectrum is obtained by time-dependent density functional theory (TD-DFT)^{58,59} calculation. Benchmark for different functionals, including PBE⁶⁰, hybrid functional with different HF compositions (B3LYP, PBE0⁶¹, M06-2X⁶²), and range-separated functional with different $\alpha/\beta/\omega$ parameters (LC-BLYP^{63,64}, CAM-B3LYP⁶⁵, ω B97XD⁶⁶), are carried out for TD-DFT calculation by comparing with the experimental data. The ω parameters in LC-BLYP are optimized to be 0.01 by using the optDFT ω package proposed by Lu Tian⁶⁷. Among these, the result of B3LYP functional is most comparable to the experimental spectra (Supplementary Fig. 40). Therefore, B3LYP functional is finally chosen in our work. Compositions of molecular orbitals are analysed based on natural atomic orbital (NAO) partition⁶⁸. All calculations are carried out in Gaussian 16⁶⁹ and Multiwfn⁷⁰ package, and the Kohn-Sham orbitals are visualized in the Visual Molecular Dynamics (VMD) program⁷¹.”

(Pages 20-21, lines 406-420)

New References:

55. Becke, A. D. A new mixing of hartree-fock and local density-functional theories. *J. Chem. Phys.* **98**, 1372-1377 (1993).
56. Weigend, F. & Ahlrichs, R. Balanced basis sets of split valence, triple zeta valence and quadruple zeta valence quality for H to Rn: design and assessment of accuracy. *Phys. Chem. Chem. Phys.* **7**, 3297-3305 (2005).
57. Grimme, S., Antony, J., Ehrlich, S. & Krieg, H. A consistent and accurate Ab initio parametrization of density functional dispersion correction (DFT-D) for the 94 elements H-Pu. *J. Chem. Phys.* **132**, 154104 (2010).
58. Runge, E. & Gross, E. K. U. Density-functional theory for time-dependent systems. *Phys. Rev. Lett.* **52**, 997-1000 (1984).
59. Van Leeuwen, R. Causality and symmetry in time-dependent density-functional theory. *Phys. Rev. Lett.* **80**, 1280-1283 (1998).

60. Perdew, J. P., Burke, K. & Ernzerhof, M. Generalized gradient approximation made simple. *Phys. Rev. Lett.* **77**, 3865-3868 (1996).
61. Adamo, C. & Barone, V. Toward reliable density functional methods without adjustable parameters: the PBE0 model. *J. Chem. Phys.* **110**, 6158-6170 (1999).
62. Zhao Y. & Truhlar, D. G. The M06 suite of density functionals for main group thermochemistry, thermochemical kinetics, noncovalent interactions, excited states, and transition elements: two new functionals and systematic testing of four M06-class functionals and 12 other functionals. *Theor. Chem. Acc.* **120**, 215-241 (2008).
63. Iikura, H., Tsuneda, T., Yanai, T. & Hirao, K. A long-range correction scheme for generalized-gradient-approximation exchange functionals. *J. Chem. Phys.* **115**, 3540-3544 (2001).
64. Baer R. & Neuhauser, D. Density functional theory with correct long-range asymptotic behavior. *Phys. Rev. Lett.* **94**, 043002 (2005).
65. Yanai, T., Tew, D. P. & Handy, N. C. A new hybrid exchange-correlation functional using the coulomb attenuating method (CAM-B3LYP). *Chem. Phys. Lett.* **393**, 51-57 (2004).
66. Chai, J.-D. & Head-Gordon, M. Long-range corrected hybrid density functionals with damped atom-atom dispersion corrections. *Phys. Chem. Chem. Phys.* **10**, 6615-6520 (2008).
67. Lu, T. optDFT ω Program v1.0, Webpage: <http://sobereva.com/346>.
68. Glendening, E. D., Landis, C. R. & Weinhold, F. Natural bond orbital methods. *wires comput. Mol. Sci* **2**, 1-42 (2012).
69. Frisch, M. J. et al. Gaussian 16, Revision A.03, Gaussian, Inc.: Wallingford, CT, (2016).
70. Lu, T. & Chen, F. Multiwfn: A multifunctional wavefunction Analyzer. *J. Comput. Chem.* **33**, 580-592 (2012).
71. Humphrey, W., Dalke, A. & Schulten, K. VMD: visual molecular dynamics. *J. Mol. Graph. Model.* **14**, 33-38 (1996).

Supplementary Figure 40. Benchmark of different functionals (PBE, B3LYP, PBE0, M06-2X, LC-BLYP/ $\omega = 0.01$, CAM-B3LYP, ω B97XD) for TD-DFT calculation of liganded $\text{Pt}_1\text{Ag}_{13}$ NC, experimental result of UV-vis spectrum is taken into comparison.

3) The authors claim that the mechanism behind DE is not demonstrated with certainty (also mentioned in Line 45 of the Introduction section). They need to improve and develop better those explanations/statements, supported by grounded experimental evidence and properly applied theoretical calculations references. Thus, the phosphorescent emission itself is explained by the authors as core-shell CT and core-core Ts. Therefore, I partially agree with the novelty regarding synthesis and experimental efforts, but not with the subsequent statement in lines 300-303 regarding DE phospho-mechanism, "which will enrich our understanding of the DE mechanism."

Response: The sentence "the mechanism behind DE is not demonstrated with certainty" has been revised as "Generally, the origin of DE in NCs can be ascribed to two first excited singlet states (S_1), S_1 and the lowest triplet excited state (T_1), and two T_1 states based on recent research on DE mechanisms of Au or AuCu NCs²⁹⁻³²." in the revised manuscript. Several references were listed to support the DE mechanism in nanoclusters.

To support the statement "which will enrich our understanding of the DE mechanism.", we performed supplementary experiments and theoretical calculations to fully elucidate the mechanism. More details of the frontier orbitals, excited states and contributions of metal and ligand fragments of $\text{Pt}_1\text{Ag}_{13}$ and $\text{Pd}_1\text{Ag}_{13}$ were given in **New Supplementary Tables 10-13**, assigning the high-, mid-, and low- energy transitions as core-shell CT states, mixed states and

core-based states, respectively. PFBT ligands were found to play an important role in the existence of DE, as its low-lying π^* levels result in energetically accessible core-shell transitions. Furthermore, a thorough PL characterization for **Pd₁Ag₁₃** NC, such as the PL lifetime data, excitation spectra, and temperature-dependent PL analysis (**New Supplementary Figures 24-28, 30, 31, 36**) were added to provide deep insight into the DE mechanism in these two structurally highly relevant NCs. Thus, the detailed phosphorescent DE mechanism was extended to open-shell structures for the first time in this work, allowing for a deeper understanding of the principles and mechanisms behind DE, which is of significant importance for the design and application of dual-emissive materials.

Revised manuscript: “Generally, the origin of DE in NCs can be ascribed to two first excited singlet states (S_1), S_1 and the lowest triplet excited state (T_1), and two T_1 states based on recent research on DE mechanisms of Au or AuCu NCs²⁹⁻³².”

(Page 3, lines 50-52)

New References:

29. Li, Q. et al. Structural distortion and electron redistribution in dual-emitting gold nanoclusters. *Nat. Commun.* **11**, 2897 (2020).

30. Luo, L., Liu, Z., Du, X. & Jin, R. Near-infrared dual emission from the Au₄₂(SR)₃₂ nanocluster and tailoring of intersystem crossing. *J. Am. Chem. Soc.* **144**, 19243-19247 (2022).

31. Arima, D., Niihori, Y. & Mitsui, M. Unravelling the origin of dual photoluminescence in Au₂Cu₆ clusters by triplet sensitization and photon upconversion. *J. Mater. Chem. C* **10**, 4597-4606 (2022).

32. Si, W.-D. et al. Two triplet emitting states in one emitter: near-infrared dual-phosphorescent Au₂₀ nanocluster. *Sci. Adv.* **9**, eadg3587 (2023).

Reviewer #2 (Remarks to the Author):

Zhu and his co-workers reported the preparation of two open-shell M_1Ag_{13} alloy nanoclusters (NCs) ($M = Pt$ and Pd) using a mixed-ligand synthetic method. While the Pt_1Ag_{13} NC exhibits interesting phosphorescent dual emission, there is no evidence that the comparable Pd_1Ag_{13} NC behaves similarly in terms of photoluminescence (PL). In Supplementary Figs. 18 and 24, the second emission peak of the Pd_1Ag_{13} NC around 830 nm is exceedingly faint and difficult to detect. Surprisingly, the authors did not provide any lifetime data, excitation spectra, or temperature-dependent PL analysis to support the claim that the Pd_1Ag_{13} NC has dual emission, nor did they look into the connection between the alloy metal cores and the PL properties of these two structurally highly relevant NCs. Due to the omission of thorough PL characterization for the Pd_1Ag_{13} NC and a lack of mechanistic insights to illustrate the PL differences between these two open-shell alloy NCs, this reviewer concludes that the quality of the current manuscript does not meet the standards of Nature Communications.

Response: Thanks for your valuable comments. The PL lifetime (**New Supplementary Figures 24, 26 and New Supplementary Table 5**) and time-dependent absorption spectra of the singlet O_2 indicator of 1, 3-diphenylisobenzofuran (DPBF) in Pd_1Ag_{13} solution (**New Supplementary Figure 28**) has been supplemented to prove that the Pd_1Ag_{13} NC has phosphorescent dual emission. Excitation spectra (**New Supplementary Figure 30**), PL lifetime under different excitation wavelength and temperature-dependent PL steady-state spectra of Pd_1Ag_{13} in 2-Me-THF was further performed and analyzed to reveal the origin of DE in Pd_1Ag_{13} (**New Supplementary Figures 24, 36**). Based on the results, we deduced that PL I may originate from the core-shell CT states, as it was more sensitive to low temperatures owing to the suppression of nonradiative relaxation processes. While PL II, which showed less dependence on temperature, may originate from core-based states. The results were closely resembled that observed in Pt_1Ag_{13} . Details were listed as follows:

(1) First, to determine the phosphorescent characteristics of Pd_1Ag_{13} , the PL decay measurement was performed. Based on the fitting results of PL I and PL II decay, the average lifetimes of Pd_1Ag_{13} were calculated to be approximately 1.034 μs and 1.509 μs , respectively. Additionally, the luminescence lifetimes of PL I and PL II were simultaneously enhanced in N_2 and simultaneously reduced in O_2 (see **New Supplementary Figure 26**). Finally, the rapid

decrease in the characteristic absorption band of the DPBF solution containing $\text{Pd}_1\text{Ag}_{13}$ confirmed the generation of singlet oxygen (see **New Supplementary Figure 28**). Further discussion has been added in the revised manuscript.

Revised manuscript: “Similar results were observed for $\text{Pd}_1\text{Ag}_{13}$, with average lifetimes of approximately 1.034 μs and 1.509 μs for PL I and PL II respectively, both exhibiting phosphorescent characteristics. (Supplementary Figs. 24-28 and Supplementary Table 5).”

(Page 8, lines 160-162)

New Supplementary Figure 26. (a) PL I and (b) PL II decay profiles of $\text{Pd}_1\text{Ag}_{13}$ nanocluster at room temperature under ambient condition (black dots), N_2 purged (red dots), and O_2 saturated (blue dots) conditions.

New Supplementary Figure 28. (a) Time-dependent absorption spectra of the 1,3-diphenylisobenzofuran (DPBF). (b) Time-dependent absorption spectra of the DPBF in solution mixed with $\text{Pd}_1\text{Ag}_{13}$ in the presence of air.

(2) Next, the PL excitation (PLE) and wavelength-dependent PL analyses was performed with $\text{Pd}_1\text{Ag}_{13}$ to reveal the origin of DE. As shown in **New Supplementary Figure 30**. The PLE

spectra of PL I and PL II have the maximum excitation bands at 420 and 600 nm, respectively. However, an additional peak located at 680 nm appeared in PLE II. Interestingly, no PL I emission was observed under 680 nm excitation (**New Supplementary Figure 31**). Furthermore, the excitation-dependent decay measurements of PL I and PL II were performed and the fitted lifetime data were summarized (**New Supplementary Table 5**). Two exponential lifetimes were required to fit the PL I and PL II decays. The decay and rise of the τ_1 and τ_2 of PL I were accompanied by the rise and decay of the τ_1 and τ_2 of PL II, similar to the results of **Pt₁Ag₁₃**. The above results indicated PL I and PL II may originate from two distinct emitting states in **Pd₁Ag₁₃**.

Revised manuscript: “The PLE and wavelength-dependent PL spectra of **Pd₁Ag₁₃** were also analyzed (Supplementary Figs. 30, 31). It was found that both PL I and PL II were present under excitation at 420, 497, and 600 nm. However, only PL II emission was observed under excitation at 680 nm. The result of excitation-dependent decay measurements of PL I and PL II in **Pd₁Ag₁₃** was similar to that observed in **Pt₁Ag₁₃** (Supplementary Table 5). Therefore, we posited that PL I and PL II in **Pd₁Ag₁₃** may also stem from two distinct emitting states.”

(Page 9, lines 182-187)

New Supplementary Figure 30. PLE spectrum of **Pd₁Ag₁₃** at PL I and PL II wavelengths.

New Supplementary Figure 31. PL spectra of $\text{Pd}_1\text{Ag}_{13}$ in 2-Me-THF under excitation wavelengths of 420, 497, 600, and 680 nm, respectively.

New Supplementary Table 5. The lifetime and relative amplitude of $\text{Pd}_1\text{Ag}_{13}$ in 2-Me-THF under different excitation wavelengths.

Ex.	PL I (748 nm)					PL II (830 nm)				
	τ_1 (μs)	τ_1 (%)	τ_2 (μs)	τ_2 (%)	τ_{av} (μs)	τ_1 (μs)	τ_1 (%)	τ_2 (μs)	τ_2 (%)	τ_{av} (μs)
420 nm	0.982	85	1.271	15	1.034	0.840	28	1.785	72	1.509
497 nm	0.774	70	1.432	30	1.059	0.718	39	1.813	61	1.592
600 nm	0.353	52	1.291	48	1.076	0.728	41	1.828	59	1.590
680 nm	0.360	53	1.308	47	1.085	0.815	49	1.853	51	1.545

(3) Finally, to understand the non-radiative relaxation processes of the two emission states of $\text{Pd}_1\text{Ag}_{13}$, temperature-dependent PL measurements were conducted. As the temperature decreased from 293 to 193 K, the emission peaks of PL I and PL II showed blue shifts of 20 nm and 10 nm, respectively (New Supplementary Figure 36a). The visual color changes of $\text{Pd}_1\text{Ag}_{13}$ with temperature and chromaticity coordinates x , y were plotted in the CIE 1931 chromaticity diagram. It was found that within this temperature range, $\text{Pd}_1\text{Ag}_{13}$ emitted red

with chromaticity coordinates of (0.71, 0.29) (New Supplementary Figure 36c). It was worth noting that PL I and PL II showed different trends with temperature changes. PL I exhibited an increasing luminescence intensity with decreasing temperature, while the luminescence intensity of PL II followed a trend of initially increasing and then decreasing (New Supplementary Figure 36b). PL I and PL II were fitted using by the Arrhenius expression and fitted Arrhenius expression to give the value of a and E_a , respectively (New Supplementary Figure 36b and New Supplementary Table 9). The temperature-dependent PL behavior of $\text{Pd}_1\text{Ag}_{13}$ highly resembled that of $\text{Pt}_1\text{Ag}_{13}$.

Revised manuscript: “The temperature-dependent steady-state PL behaviour of $\text{Pd}_1\text{Ag}_{13}$ in 2-Me-THF closely resembled that observed in $\text{Pt}_1\text{Ag}_{13}$. The intensity of PL I was increased by 2.88 times, along with a 20 nm blue shift, while the intensity of PL II reached its maximum at 223 K with a 10 nm blue shift, as the temperature decreased. (Supplementary Fig. 36a, b and Supplementary Table 8). The CIE coordinates of (0.71, 0.29) were identical upon temperature decreasing (Supplementary Fig. 36c). The calculated E_a values was 115.73 meV and the a value of $\text{Pd}_1\text{Ag}_{13}$ also showed a sharp decrease from 115.73 (PL I) to 58.27 and -10.43 (PL II a_1 and a_2 , Supplementary Table 9). Hence, we concluded that in both $\text{Pt}_1\text{Ag}_{13}$ and $\text{Pd}_1\text{Ag}_{13}$, PL I may originate from the core-shell CT states, as it was more sensitive to low temperatures owing to the suppression of nonradiative relaxation processes. While PL II, which showed less dependence on temperature, may originate from core-based states.”

(Page 12, lines 235-244)

New Supplementary Figure 36. (a) Variable-temperature PL spectra of $\text{Pd}_1\text{Ag}_{13}$ in 2-Me-THF. (b) Normalized integrated PL I and PL II intensities were fitted using *Eqs 1* and *2*, respectively; the integration of PL II is separated as regions I and II. (c) CIE 1931 color space chromaticity

diagram showing the luminescence color change of **Pd₁Ag₁₃** in the temperature range of 193-293 K.

New Supplementary Table 9. Fitting parameter obtained by fitting the temperature dependence of the PL I and PL II intensities of **Pd₁Ag₁₃** with Arrhenius equations, taking into account one (PL I) and two (PL II) non-radiative channels.

Peak	a₁	Ea₁(meV)	a₂	Ea₂(meV)
PL I	115.73	20.81	NA	NA
PL II	58.27	136.62	-10.43	19.28

In addition to the serious issues mentioned above, the authors should take the following into consideration:

1. Generally speaking, 7-electron open-shell NCs have poor stability. The authors must offer evidence to show that the two NCs remain unchanged during the PL measurements in 2-methyltetrahydrofuran. The exploration of dual emission in other organic solvents is highly recommended.

Response: The time-tracking UV-vis absorption spectra and PL spectra of **Pt₁Ag₁₃** and **Pd₁Ag₁₃** in 2-Me-THF were performed. As shown in **New Supplementary Figures 32, 33**, the position and intensity of the characteristic absorption peak and the emission peak showed no change within 5 hours, confirming the stability of the two NCs during PL measurements.

The dual emission of **Pt₁Ag₁₃** and **Pd₁Ag₁₃** in other organic solvents was characterized (**Response Figures 1, 2**). Both PL I and PL II appeared in different organic solvents. Interestingly, with the decrease in solvent polarity from the highly polar 2-Me-THF to the less polar toluene, both PL I and PL II intensities increased to different extents. This result was consistent with the findings of Jin and Sun (Nat. Commun. 2020, 11, 2897; Sci. Adv. 2023, 9, eadg3587, etc.).

Supplementary Figure 32. Time-tracking (a) UV-vis and (b) PL spectra of $\text{Pt}_1\text{Ag}_{13}$ in 2-Me-THF solution at room temperature.

New Supplementary Figure 33. Time-tracking (a) UV-vis and (b) PL spectra of $\text{Pd}_1\text{Ag}_{13}$ in 2-Me-THF solution at room temperature.

Response Figure 1. Solvent polarity dependence PL spectra of $\text{Pt}_1\text{Ag}_{13}$ at the excitation of 390 nm.

Response Figure 2. Solvent polarity dependence PL spectra of **Pd₁Ag₁₃** at the excitation of 420 nm.

2. It is insufficient to investigate the origin of dual emission just using DFT calculations. As the most important part of the current work, the authors should carry out additional experimental studies to obtain comprehensive information regarding the photoexcited species of the **M₁Ag₁₃** NCs. Transient absorption measurements along with the corresponding analysis are necessary (see Nat. Commun. 2020, 11, 2897; J. Am. Chem. Soc. 2022, 144, 19243; Sci. Adv. 2023, 9, eadg3587, etc.)

Response: The femtosecond (fs) and ns-TA spectroscopy were performed on **Pt₁Ag₁₃** to probe the excited-state dynamics of two emitting states. Further discussion has been supplemented in the revised manuscript.

Revised manuscript: “To understand the photophysics of DE, we performed time-resolved transient absorption (TA) spectroscopy measurements with **Pt₁Ag₁₃**. We first looked into the nanosecond relaxation dynamics of **Pt₁Ag₁₃** by performing ns-TA with excitation of 400 nm and 600 nm. Fig. 4a showed the ns-TA data map with excitation of 400 nm that consisted of a negative band at 475 nm and a positive band across 520 nm to 900 nm. The negative band could be assigned to the ground state bleach (GSB) signal which coincided with the UV-vis absorption spectrum as shown in Fig. 4a, b, and the positive band was the excited state absorption (ESA) of the triplet state. The ns-TA data presented a monotonous decay without spectral shift, suggesting no new transient species were generated. The ns-TA data map (Fig. 4b) under 600 nm excitation was similar to that excited at 400 nm, which may be because the excited state dynamics of **Pt₁Ag₁₃** under the different-energy excited laser were very close to each other thus

making the ns-TA set-up cannot distinguish the differences. This was further demonstrated by the almost overlapped kinetic traces at 560 nm with an average lifetime of less than 1 μ s ($t_1 = 71$ ns, $t_2 = 625$ ns, Fig. 4c), which was close to the lifetime obtained from the fluorescence lifetime (around 1 μ s). We also conducted the fs-TA measurements under 400 nm excitation, the kinetic traces at 600 nm with a lifetime larger than 2 ns were displayed in Fig. 4d, and no more new transient components were obtained. These results indicated that TA spectroscopy mainly probed the dynamics of core-shell CT excited state (PL I), which was much stronger than core-based one (PL II). These results were consistent with the ns-TA test results of reported Au₂₀³².”

Fig. 4: Excited-state dynamics of Pt₁Ag₁₃.

a-b The ns-TA data of Pt₁Ag₁₃ under 400 nm (a) excitation and 600 nm (b) excitation, all time-resolved spectroscopy measurements are conducted with nitrogen protection. **c** The kinetic traces at 560 nm of Pt₁Ag₁₃ under 400 nm excitation and 600 nm excitation. **d** The kinetic trace at 600 nm extracted from the fs-TA data.

(Pages 12-14, lines 245-267)

New references:

32. Si, W.-D. et al. Two triplet emitting states in one emitter: near-infrared dual-phosphorescent Au₂₀ nanocluster. *Sci. Adv.* **9**, eadg3587 (2023).

3. A highly relevant reference on dual-emission mechanisms is missing (see *J. Phys. Chem. Lett.* 2021, 12, 1514).

Response: Thanks for your careful review. The relevant reference has been added in the revised manuscript.

New References:

23. Zhou, M. & Song, Y. Origins of visible and near-infrared emissions in [Au₂₅(SR)₁₈]⁻ nanoclusters. *J. Phys. Chem. Lett.* **12**, 1514-1519 (2021).

24. Lee, D. et al. Electrochemistry and optical absorbance and luminescence of molecule-like Au₃₈ nanoparticles. *J. Am. Chem. Soc.* **126**, 6193-6199 (2004).

25. Link, S. et al. Visible to infrared luminescence from a 28-atom gold cluster. *J. Phys. Chem. B* **106**, 3410-3415 (2002).

(Page 24, lines 477-482)

Reviewer #3 (Remarks to the Author):

Although the dual-emission (DE) of metal nanoclusters has been widely studied due to its promising applications in ratio metric sensing, bioimaging, novel optoelectronic devices, etc., there are rare examples of DE with open-shell configurations. Moreover, the lack of fundamental understanding between structure and DE properties in open-shell NCs with doublet or quartet emissions impedes exploring the controllable synthesis of NCs with DE. This work reports the synthesis and photophysical properties of dual-emissive nanoclusters $M_1Ag_{13}(PFBT)_6(TPP)_7$ (M_1Ag_{13} ; $M = Pt, Pd$) with open shell electronic structure, and clarifies the DE mechanism attributed to quartet states which involves core-shell CT and core-based excitation, based on time-resolved spectroscopy and theoretical calculations. This paper reports a novel point about dual-quartet phosphorescent emission in the open-shell Pt_1Ag_{13} nanocluster. Thus, I strongly recommend publishing this interesting work in Nature Communications.

Minor revisions:

1. A closed-shell electronic configuration is always preferred for clusters of group 11 elements, and superatom theory and electron-counting rules have been developed to explain cluster stability. So how is the stability of Pt_1Ag_{13} and Pd_1Ag_{13} NCs with one unpaired electron? If stable, how can they be stabilized?

Response: Many thanks for your careful review. The time-tracking UV-vis absorption spectra and PL spectra of Pt_1Ag_{13} and Pd_1Ag_{13} in 2-methyltetrahydrofuran were performed. As shown in **New Supplementary Figures 32, 33**, the position and intensity of the characteristic absorption peak and the emission peak showed no change within 5 hours, confirming the stability of the two NCs during PL measurements. The stability of Pt_1Ag_{13} and Pd_1Ag_{13} may be due to abundant noncovalent interactions within and between single NC. The C-H \cdots F, C-F \cdots π and $\pi\cdots\pi$ interactions within and between each NC increased the rigidity and stability of the Pt_1Ag_{13} and Pd_1Ag_{13} and facilitated crystallization. (**Response Figure 3**).

Supplementary Figure 32. Time-tracking (a) UV-vis and (b) PL spectra of $\text{Pt}_1\text{Ag}_{13}$ in 2-Me-THF solution at room temperature.

New Supplementary Figure 33. Time-tracking (a) UV-vis and (b) PL spectra of $\text{Pd}_1\text{Ag}_{13}$ in 2-Me-THF solution at room temperature.

Response Figure 3. (a) The intracluster $\text{C-H}\cdots\text{F}$, $\text{C-F}\cdots\pi$ and $\pi\cdots\pi$ interactions in M_1Ag_{13} . (b) Intercluster $\text{C-H}\cdots\text{F}$, $\text{C-F}\cdots\pi$ and $\pi\cdots\pi$ interactions in the unit cell of M_1Ag_{13} . The

blue/red/green dashed lines correspond to the C-H \cdots F, C-F \cdots π and $\pi\cdots\pi$ interactions, respectively. Color labels: yellow, Pt/Pd; sky blue, Ag; red, S; magenta, P; orange, F; gray, C.

2. The authors mention that PL I is centered at 650 nm while PL II is centered at 825 nm in Pt₁Ag₁₃, and further assign PL I to the core-shell charge transfer states and PL II to the core states. My question is how can the author make sure the DE might not come from the cluster itself but from some impurity or decompositions? To answer this question, it would be useful to show the ¹H NMR spectra of Pt₁Ag₁₃.

Response: To ensure that DE comes from the cluster itself rather than other impurities or decomposition products, we performed ESI-MS, ¹H NMR and stability tests. In the ESI-MS spectrum of Pt₁Ag₁₃, a strong signal was observed at m/z 4761.07 (**Figure 1c**), corresponding to the [Pt₁Ag₁₃(PFBT)₆(TPP)₇Cs]⁺ species (calculated at m/z 4761.06). Furthermore, the ¹H NMR showed a peak at 7-8 ppm belonging to the hydrogen atoms in PPh₃ ligands, with the other signals mainly coming from the solvents (**Response Figure 4**). Finally, the time-tracking UV-vis absorption spectra and PL spectra of Pt₁Ag₁₃ and Pd₁Ag₁₃ in 2-methyltetrahydrofuran were performed, confirming the stability of the two NCs during PL measurements within 5 hours (**Supplementary Figure 32**).

Figure 1c. ESI-MS results of Pt₁Ag₁₃ nanocluster. Insets: experimental (in black) and simulated (in red) isotope patterns.

Response Figure 4. ^1H NMR spectra measured by $\text{Pt}_1\text{Ag}_{13}$ using NaBH_4 as a reducing agent (in CD_2Cl_2).

Supplementary Figure 32. Time-tracking (a) UV-vis and (b) PL spectra of $\text{Pt}_1\text{Ag}_{13}$ in 2-Me-THF solution at room temperature.

3. The authors confirm the singlet oxygen production by decreasing the characteristic absorption peak of 1,3-diphenylisobenzofuran, indicating that $\text{Pt}_1\text{Ag}_{13}$ is phosphorescent. The singlet oxygen experiment in $\text{Pd}_1\text{Ag}_{13}$ should also be supplemented to identify the PL properties.

Response: The time-dependent absorption spectra of the singlet O_2 indicator of 1, 3-diphenylisobenzofuran (DPBF) in $\text{Pd}_1\text{Ag}_{13}$ solution (**New Supplementary Figure 28**) has been supplemented to prove that the $\text{Pd}_1\text{Ag}_{13}$ NC has phosphorescent dual emission.

New Supplementary Figure 28. (a) Time-dependent absorption spectra of the 1, 3-diphenylisobenzofuran (DPBF). (b) Time-dependent absorption spectra of the DPBF in solution mixed with $\text{Pd}_1\text{Ag}_{13}$ in the presence of air.

4. It seems the authors take some simplifications for ligands in theoretical calculation. Does it have any influence on PL characters?

Response: Many thanks for the valuable suggestion. There are two kinds of organic ligands in this ligand-protected $\text{Pt}_1\text{Ag}_{13}$ cluster, PFBT (pentafluorobenzenethiol) and TPP (triphenylphosphine). In the first simplified structure (G1), we use methyl groups to replace benzene in TPP ligands. In the second simplified structure (G2), we use methyl groups to replace benzene in TPP and fluorobenzene groups of PFBT. We compare the HOMO-LUMO energy gaps and orbitals energy levels of the original structure and two simplified structures. The results reveal that methyl substitution of benzene groups in TPP (G1) have little influence on its orbital character, while replacement of fluorobenzene groups (G2) obviously change the orbital energy for the liganded $\text{Pt}_1\text{Ag}_{13}$ cluster (**Response Figure 5**). Moreover, details of the excited states in **Supplementary Tables 10-13** reveals that TPP ligands are almost not involved in the transitions. Therefore, in this study, we use methyl to replace benzene groups, in order to simplify the structure and reduce the computational cost of the theoretical investigation.

Response Figure 5. Comparison of the orbital energy levels for original and different simplified structures of liganded $\text{Pt}_1\text{Ag}_{13}$ NC. (G0-Full Original structure; G1-Benzene groups in TPP ligands are replaced by methyl groups; G2-Benzene in TPP ligands and fluorobenzene groups of PFBT are replaced by methyl groups)

5. It is recommended to explain the chromaticity coordinates x and y in Fig. 3C and to describe how to make this plot.

Response: In the CIE diagram, the chromaticity coordinates x and y are used to describe the position of a color in the chromaticity diagram, and further to describe the hue and saturation of the color, which can help us understand and analyze the characteristics of colors. The chromaticity coordinates x and y are obtained by normalizing the colors in the CIE chromaticity diagram. The values of chromaticity coordinates x and y range from 0 to 1, where the sum of x and y must be less than or equal to 1.

Revised manuscript: “The visualized color change of $\text{Pt}_1\text{Ag}_{13}$ with temperature and the chromaticity coordinates x and y were plotted on a CIE 1931 color space chromaticity diagram, showing a shift from reddish-orange (CIE: 0.65, 0.35) to orange (CIE: 0.61, 0.39), as shown in Fig. 3c.”

(Page 10, lines 197-200)

6. The yields of $\text{Pt}_1\text{Ag}_{13}$ and $\text{Pd}_1\text{Ag}_{13}$ NCs were not given in the SI.

Response: The yields of $\text{Pt}_1\text{Ag}_{13}$ and $\text{Pd}_1\text{Ag}_{13}$ were about 11.5% and 10.3%, respectively, based on the Ag element. The data have been supplemented in the revised manuscript.

Revised manuscript: “The yield was 11.5% based on the Ag element (calculated from the AgNO_3) for synthesizing the $\text{Pt}_1\text{Ag}_{13}$ nanocluster.”

“The yield was 10.3% based on the Ag element (calculated from the AgNO_3) for synthesizing the $\text{Pd}_1\text{Ag}_{13}$ nanocluster.”

(Page 19, lines 374-380)

7. The format of several references is incorrect (i.e., the capital letter, missing subscript, Italic, etc.).

Response: Thank you for your careful review. The format of references have been thoroughly checked and corrected.

New References:

6. Wang, J.-J. et al. High efficiency warm-white light-emitting diodes based on copper-iodide clusters. *Nat. Photon.* **18**, 200-206 (2024).

14. Lou, X. et al. Surface motif sensitivity of dual emissive gold nanoclusters for robust ratiometric intracellular imaging. *Chem. Commun.* **56**, 7112-7115 (2020).

18. Kawagoe, R., Takashima, I., Uchinomiya, S. & Ojida, A. Reversible ratiometric detection of highly reactive hydropersulfides using a FRET-based dual emission fluorescent probe. *Chem. Sci.* **8**, 1134-1140 (2017).

(Pages 22-23, lines 437-466)

Reviewer #1 (Remarks to the Author):

The revised manuscript titled "Dual-Quartet Phosphorescent Emission in Open-Shell M1Ag13 (M = Pt, Pd) Nanoclusters" has undergone significant and crucial revisions to meet the standards of Nature Communications.

From a theoretical standpoint, the authors have transparently addressed all completed inquiries related to DFT and TDDFT calculations. Additionally, the inclusion of further theoretical calculations and their discussion has enhanced the quality of the research. Consequently, crucial experimental inquiries have been thoroughly addressed and correlated with the calculations. In its revised form, the manuscript demonstrates a superior level of research excellence.

Therefore, I highly recommend its publication on Nature Communications in its revised form.

Reviewer #2 (Remarks to the Author):

This reviewer is satisfied with the current revision to the previous concerns. There is one thing that has to be clarified before publication:

The authors stated in the revised manuscript: "This DE character of Pt1Ag13 and Pd1Ag13 NC largely depends on the role of electronegative PFBT ligands, providing an effective blueprint for designing materials with dual emissions." However, the photo-luminescent quantum yields (QYs) for Pt1Ag13 and Pd1Ag13 were estimated to be 1.49% and 0.07%, respectively. The authors should provide an explanation for the significant QY difference.

Reviewer #3 (Remarks to the Author):

All the issues have been addressed. The manuscript can be accepted.

We thank all reviewers for their helpful comments. The response to reviewer 2 is shown in blue in this letter, and revisions are in red.

Reviewer #2 (Remarks to the Author):

This reviewer is satisfied with the current revision to the previous concerns. There is one thing that has to be clarified before publication:

The authors stated in the revised manuscript: “This DE character of Pt₁Ag₁₃ and Pd₁Ag₁₃ NC largely depends on the role of electronegative PFBT ligands, providing an effective blueprint for designing materials with dual emissions.” However, the photoluminescent quantum yields (QYs) for Pt₁Ag₁₃ and Pd₁Ag₁₃ were estimated to be 1.49% and 0.07%, respectively. The authors should provide an explanation for the significant QY difference.

Response: The difference in the photoluminescence quantum yield (PLQY) between Pt₁Ag₁₃ and Pd₁Ag₁₃ NCs may be due to different electron affinity of Pt or Pd. The stronger electron affinity of Pt may lead to enhanced charge transfer abilities, resulting in higher PLQY in Pt₁Ag₁₃.

Revised Manuscript: “The difference in the PLQY between the two NCs may be caused by different electron affinity of Pt or Pd (i.e., the capability to attracting electron)^{45,46}. The stronger electron affinity of Pt may lead to enhanced charge transfer abilities, resulting in higher PLQY in Pt₁Ag₁₃.”

(Pages 6-7, lines 138-141)

New References:

45. Ho, J. et al. A study of the electronic structures of Pd₂⁻ and Pd₂ by photoelectron spectroscopy. *J. Chem. Phys.* **95**, 4845-4853 (1991).

46. Hotop, H. & Lineberger, W. C. Binding energies in atomic negative ions: II. *J. Phys. Chem. Ref. Data* **14**, 731-750 (1985).